

# Real-time Tsunami Force Prediction by Mode Decomposition -Based Surrogate Modeling

Kenta Tozato[1], Shinsuke Takase[2], Shuji Moriguchi[3], Kenjiro Terada[3], Yu Otake[4], Yo Fukutani[5], Kazuya Nojima[6], Masaaki Sakuraba[6], and Hiromu Yokosu[7]

[1]Department of Civil and Environmental Engineering, Tohoku University, Aza-Aoba 468-1, Aramaki, Aoba-ku, Sendai 980-8572, Japan

[2]Department of Civil Engineering and Architecture, Hachinohe Institute of Technology, 88-1 Ohbiraki, Myo, Hachinohe, Aomori 031-8501, Japan

[3]International Research Institute of Disaster Science, Tohoku University, Aza-Aoba 468-1, Aramaki, Aoba-ku, Sendai 980-8572, Japan

[4]Department of Civil and Environmental Engineering, Tohoku University, Aza-Aoba 6-6-01, Aramaki, Aoba-ku, Sendai 980-8579, Japan

[5]College of Science and Engineering, Kanto Gakuin University, Mutsuura Higashi 1-50-1, Kanazawa-ku, Yokohama-shi, Kanagawa 236-8501, Japan

[6]Research and Development Center, Nippon Koei Co., Ltd. Inarihara, 2304, Tsukuba-shi, Ibaraki 300-1259, Japan

[7]Nuclear Safety Research  Development Center, Chubu Electric Power Co., Inc., Sakura 5561, Omaezaki, Shizuoka 437-1695, Japan

Correspondence: Shuji Moriguchi (s_mori@irides.tohoku.ac.jp)

**Abstract.** This study presents a framework for real-time tsunami force predictions by the application of mode decomposition based surrogate modelling with 2D-3D coupled numerical simulations. A limited number of large-scale numerical analyses are performed for a selection scenarios with variations in fault parameters to capture the distribution tendencies of the target risk indicators. Then, the proper orthogonal decomposition (POD) is applied to the analysis results to extract the principal

5    modes that represent the temporal and spatial characteristics of tsunami forces. A surrogate model is then constructed by a linear combination of these modes, whose coefficients are defined as functions of the selected input parameters. A numerical example is presented to demonstrate the applicability of the proposed framework to one of the tsunami-affected areas during the Great East Japan Earthquake of 2011. Combining 2D and 3D versions of the stabilized finite element method, we carry out a series of high precision numerical analyses with different input parameters to obtain a set of time history data of the

10    tsunami forces acting on buildings and the inundation depths. POD is applied to the data set to construct the surrogate model that is capable of providing the predictions equivalent to the simulation results almost instantaneously. Based on the acceptable accuracy of the obtained results, it was confirmed that the proposed framework is a useful tool for evaluating time series data of hydrodynamic force acting on buildings.

## 1    Introduction

15    In order to estimate the potential damage due to a tsunami, predictions need to consider both the global aspect, such as the scale of the inundation areas, and also the local hydrodynamic forces acting on individual houses and each type of infrastructure. In



fact, the potential effects of tsunami force have been implemented in recent design standards (ASCE, 2017; Nakano, 2017). In addition, thanks to the extensive efforts made in recent years to develop numerical analysis techniques, maturity can be seen especially in the numerical analysis techniques applied to coastal engineering which provide highly accurate evaluations and predictions of tsunami forces (e.g., Qin et al., 2018; Xiong et al., 2019).

It is also extremely important for adequate disaster responses, such as evacuation actions, that information about the extent of damage can be quickly ascertained. In response to this demand, numerous studies have been made on instantaneous tsunami predictions. For example, NOAA (National Oceanic and Atmospheric Administration) constructed a real-time prediction system for tsunamis based on a long-term observation database (e.g., Titov et al., 2005; Tang et al., 2008; Wei et al., 2008; Tang et al., 2009). In addition, there have been many studies focused on the development of techniques for making real-time damage predictions based on numerical simulations. The studies can be classified into two types. In the first approach, the source and seismic waveform data are obtained from real-time observation systems, and then the tsunami propagation is calculated using input data estimated from the observed data. There have been a number studies related to this approach; one on data assimilation-based tsunami forecasting (Maeda et al., 2015), another on tsunami forecasting based on inversion for initial sea surface height (Tsushima et al., 2014), another to predict tsunami waveform using extreme learning machine (Mulia et al., 2016) and another using a real-time inundation simulation with supercomputers (e.g., Oishi et al., 2015; Musa et al., 2018). The second approach is based on the use of precomputed simulation data. Numerical simulations with multiple scenarios are carried out and tsunami database is numerically created to output prediction results immediately after an earthquake occurs. This approach is also widely accepted and several studies have been reported. Among these is a study on making real-time predictions using source information and precomputed tsunami waveform and inundation databases (Gusman et al., 2014), another uses a combination of precomputed tsunami databases and the neural networks (Fauzi and Mizutani, 2019), and another features the surrogate modeling of numerical simulation results (e.g., Fukutani et al., 2019; Kotani et al., 2020).

The objective of this study is to evaluate the time-series of the spatial distribution of tsunami forces acting on buildings in real time. Because a very high computational cost is required to evaluate the tsunami force over a wide area, it is generally difficult to perform such simulations in real time. Therefore, a surrogate modeling-based prediction method is employed in this study. However, since this study considers multiple evaluation points to determine tsunami force, the methods employed in Fukutani et al. (2019) and Kotani et al. (2020), which require surrogate models to be defined at each evaluation point, are inefficient. To overcome this problem, a mode decomposition technique is used to construct a surrogate model. Spatial modes and their equivalent principal components have been successfully utilized to construct surrogate models in the research fields of earthquake engineering and tsunami engineering. My Ha et al. (2008) proposed a surrogate model of a tsunami simulation by means of Proper Orthogonal Decomposition (POD) for the purpose of reducing the dimensionality or, equivalently, the computational costs. LeVeque et al. (2016) and Melgar et al. (2016) utilize the Karhunen–Loève expansion to consider the distribution of slips on a fault under various scenarios. In the fields of earthquake and structural engineering, there have been similar studies. Nojima et al. (2018) proposed to combine numerical simulations and a mode decomposition technique that is based on the singular value decomposition for predicting the distribution of strong ground motions. Bamer and Bucher (2012) also used POD to construct a surrogate 50 model to predict structural behaviour using a non-linear finite element analysis





method. Since POD enables us to systematically extract the feature quantities of the temporal and spatial distributions of risk indicators, it is considered suitable for surrogate modelling for evaluating all kinds of disaster risks. Although some studies using mode decomposition techniques have been reported in the research field of disaster science, the time-space surrogate 
modeling of tsunami force based on a high precision 3D tsunami simulation has not been extensively studied.

The present study proposes a framework for real-time tsunami force predictions by the application of the POD-based surrogate modelling of numerical simulations. Only a limited number of large-scale numerical analyses are performed for a selection of scenarios with various fault parameters to capture the distribution tendencies of the target risk indicators. After 2D shallow water simulations are performed, the results are used as input data for 3D flow simulations to evaluate the time histories of 
forces acting on buildings. Then, POD is applied to the evaluation results to extract the principal modes that represent the temporal and spatial characteristics of the forces caused by a target tsunami. A surrogate model can then be constructed by a linear combination of spatial modes whose coefficients are determined by means of the regression analysis and interpolation techniques. To demonstrate the applicability of the proposed framework, one of the tsunami-affected areas during the Great East Japan Earthquake of 2011 is targeted. Using the 2D simulation and the results are obtained for different input parameters, we 
carry out a series of 3D numerical analyses to obtain a set of time history data of the spatial distributions of tsunami force and then construct a surrogate model capable of predicting the time variations of the spatial distributions of tsunami forces almost instantaneously. A surrogate model of the inundation depth is also constructed and compared with that of tsunami force.

The structure of this paper is as follows. Section 2 explains a flow and specific procedures of the proposed framework. In Section 3, the proposed framework is applied to a target city to construct the surrogate model that enables real-time predictions 
of tsunami forces equivalent to the 3D tsunami runup simulations.

## 2 A proposed framework for the real-time tsunami force prediction

This section describes a flow and methodologies of the proposed framework. As mentioned in previous section, principal spatial modes are extracted from precomputed simulation data that are obtained from 2D-3D coupled tsunami analysis, and the surrogate models are constructed using the spatial modes for the real-time tsunami force prediction. Fig.1 shows the flowchart 
of the proposed method. Each parts of the proposed method, such as tsunami analyses, mode decomposition, and surrogate modeling, are explained in details in the following subsections.

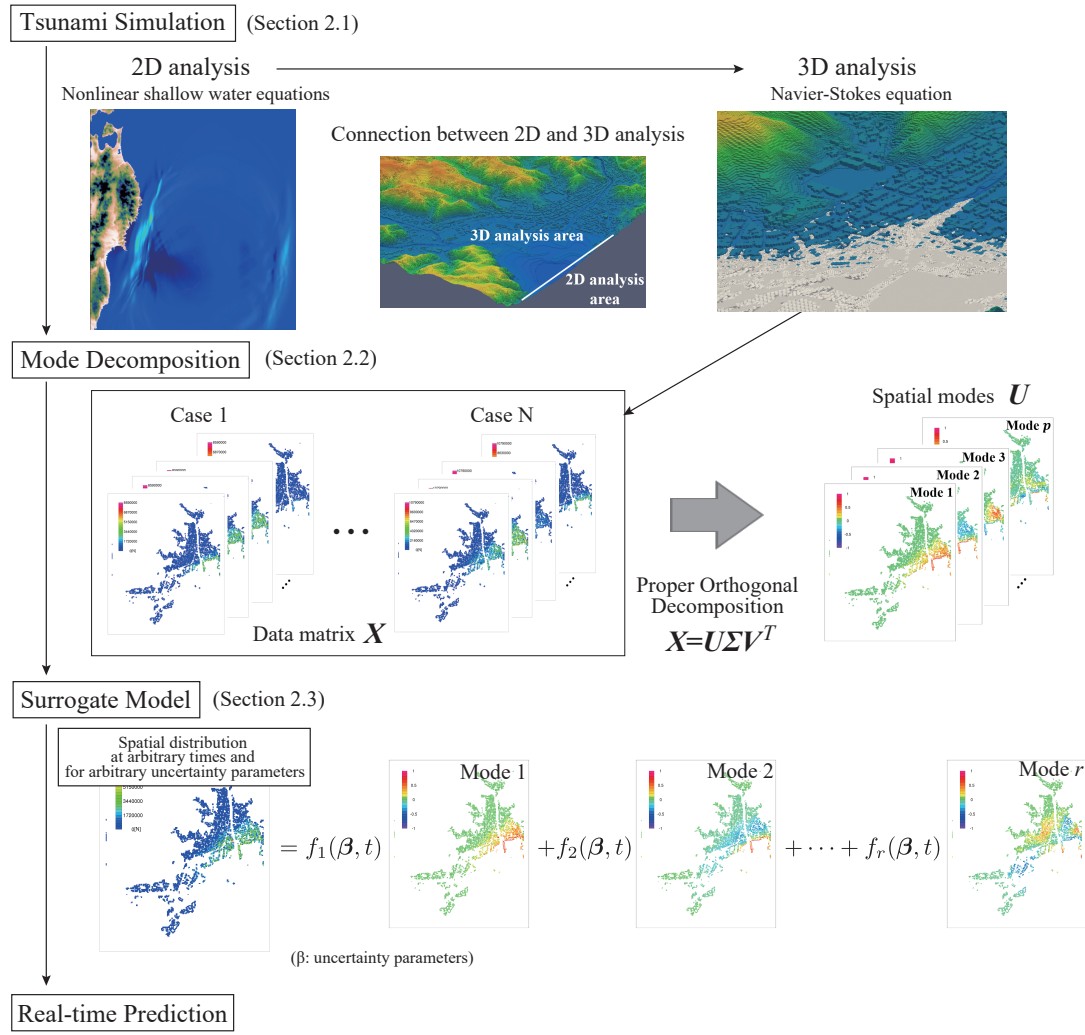

**Figure 1.** Flowchart of the real-time tsunami force prediction by mode decomposition-based surrogate model.

## 2.1 2D-3D coupled tsunami analyses

To collect the data necessary for the surrogate modelling explained in the previous section, a set of two sequential numerical analyses is carried out for each case with a selected set of fault parameters. A 2D tsunami analysis is first carried out for to obtain the information about the tsunami wave caused by the offshore fault. Then, the time histories of the tsunami height and flow velocity on the specified boundary of the target urban area are extracted for use as input data for a 3D tsunami runup simulation. A 3D numerical analysis is performed to elaborately evaluate the spatial and temporal distributions of risk indicators within the target area. Since the risk indicators in this study include not only the inundation depth, but also the force





acting on buildings, the 3D calculation requires a high-fidelity numerical analysis. In this section, only the governing equations
for 2D and 3D tsunami simulations are outlined. For the details of their connection, Takase et al. (2016) can be referenced.

To numerically analyze tsunami wave propagation over a wide area, 2D shallow water simulations are commonly performed.
TUNAMI-N2 (Imamura (1995); Goto et al. (1997)), one of the most well-known simulation codes based, is based on the
concept of the shallow water theory. In this study, we employ the TUNAMI-N2 to represent wave propagation in the offshore
area. The following formats of continuity and nonlinear long wave equations are solved in the 2D analysis:

$$\frac{\partial \eta}{\partial t} + \frac{\partial M}{\partial x} + \frac{\partial N}{\partial y} = 0 \tag{1}$$

$$\frac{\partial M}{\partial t} + \frac{\partial}{\partial x}\left[\frac{M^2}{D}\right] + \frac{\partial}{\partial y}\left[\frac{MN}{D}\right] + gD\frac{\partial \eta}{\partial x} + \frac{gn^2}{D^{\frac{7}{3}}}N\sqrt{M^2+N^2} = 0 \tag{2}$$

$$\frac{\partial N}{\partial t} + \frac{\partial}{\partial x}\left[\frac{MN}{D}\right] + \frac{\partial}{\partial y}\left[\frac{N^2}{D}\right] + gD\frac{\partial \eta}{\partial y} + \frac{gn^2}{D^{\frac{7}{3}}}N\sqrt{M^2+N^2} = 0 \tag{3}$$

where $M$ and $N$ are the flow rates in the $x$ and $y$ directions, $\eta$ is the water level, $D$ is the total water depth, $g$ is gravitational
acceleration and $n$ is the Manning roughness coefficient. To evaluate the force acting on buildings, we employ the following
set of 3D Navier-Stokes and continuity equations in the analysis domain $\Omega_{ns} \in R^3$

$$\rho\left(\frac{\partial \boldsymbol{u}}{\partial t} + \boldsymbol{u}\cdot\nabla\boldsymbol{u} - \boldsymbol{f}\right) - \nabla\cdot\boldsymbol{\sigma} = 0 \tag{4}$$

$$\nabla\cdot\boldsymbol{u} = 0 \tag{5}$$

where $\rho$ is the mass density, $\boldsymbol{u}$ is the velocity vector, $\boldsymbol{\sigma}$ is the stress tensor, and $\boldsymbol{f}$ is the body force vector. Also, assuming a
Newtonian fluid, the stress is calculated as

$$\boldsymbol{\sigma} = -p\boldsymbol{I} + 2\mu\varepsilon(\boldsymbol{u}) \tag{6}$$

where $p$ is the pressure, $\mu$ is the coefficient of viscosity, and $\varepsilon(\boldsymbol{u})$ is the velocity gradient tensor defined as

$$\varepsilon(\boldsymbol{u}) = \frac{1}{2}\left(\nabla\boldsymbol{u} + (\nabla\boldsymbol{u})^T\right) \tag{7}$$

To solve the governing equations of the 3D analysis, the stabilized finite element method(SFEM) is employed in this study.
Details of the method is explained in the appendix(see Appendix A).

## 2.2 Mode Decomposition

Proper orthogonal decomposition (POD) (Liang et al. (2002)) is well known as a technique to extract the principal direction
along which the variance of the collected data is maximized. In other words, POD is capable of grasping the characteristics



of data and expressing data in a lower dimension. When applying POD to data, we prepare a matrix storing the data arranged in accordance with a certain rule (hereafter, referred to as the "data matrix"). When $n$ data are obtained for scenario case $i$, they are stored into a column vector denoted by $\boldsymbol{x}_i$, which has n components. When the scenario ranges from 1 to $N$, the data matrix can be defined as

$$\boldsymbol{X} = \begin{pmatrix} | & & | \\ \boldsymbol{x}_1 & \cdots & \boldsymbol{x}_N \\ | & & | \end{pmatrix} \tag{8}$$

Here, a vertical line is added to each column vector to implicate that the component alignment follows the specified rule. The covariance matrix of the data matrix is defined as

$$\boldsymbol{C} = \boldsymbol{X}\boldsymbol{X}^T \tag{9}$$

The eigenvectors are the principal directions, and the corresponding eigenvalue represents the variance in each principal direction. Also, larger eigenvalues have more major contributions to the data.

Let $\lambda_j$ be the $j$-th eigenvalue of $\boldsymbol{C}$, and $\boldsymbol{u}_j$ be the $j$-th eigenvector ($j = 1, ..., n$). Since each eigenvalue corresponds to the variance in the corresponding principal direction or mode, those with small eigenvalues have almost no significant values and provide little meaningful information, and therefore can be omitted in the sequel. To omit unnecessary modes, the contribution rate is generally introduced as an indicator to compare the magnitude of each eigenvalue with others. Specifically, the contribution rate $d_j$ of a particular mode $j$ is defined as a percentage of each eigenvalue against the sum of all eigenvalues as

$$d_j = \frac{\lambda_j}{\sum_{k=1}^{n} \lambda_k} \tag{10}$$

When this rate is arranged in descending order, the threshold for the omission is determined according to the profile of its decreasing curve. It should be noted that the information about the omitted mode is lost and is a source of error.

Also, based on the theory of singular value decomposition, data matrix $\boldsymbol{X}$ can be expressed as

$$\boldsymbol{X} = \begin{pmatrix} | & & | \\ \boldsymbol{x}_1 & \cdots & \boldsymbol{x}_N \\ | & & | \end{pmatrix} = \boldsymbol{U}\boldsymbol{\Sigma}\boldsymbol{V}^T = \begin{pmatrix} | & & | \\ \boldsymbol{u}_1 & \cdots & \boldsymbol{u}_p \\ | & & | \end{pmatrix} \begin{pmatrix} \sqrt{\lambda_1} & & \\ & \ddots & \\ & & \sqrt{\lambda_p} \end{pmatrix} \begin{pmatrix} - & \boldsymbol{v}_1 & - \\ & \vdots & \\ - & \boldsymbol{v}_p & - \end{pmatrix} \tag{11}$$

where $\boldsymbol{U}$ is a matrix lining up the eigenvector $\boldsymbol{u}_j$ of C in the column direction and $\boldsymbol{V}$ is the matrix lining up the eigenvector $\boldsymbol{v}_j$ and matrix $\boldsymbol{C}' = \boldsymbol{X}^T\boldsymbol{X}$ in the column direction. Also, $\boldsymbol{\Sigma}$ is the matrix having the square roots of eigenvalues in the diagonal components, which are referred to as singular values. Here, $p$ is the number of singular values that must be greater than zero. Equation (11) can be substituted for $\boldsymbol{C}$ and $\boldsymbol{C}'$ to have their eigenvalue decomposition as

$$\boldsymbol{C} = \boldsymbol{X}\boldsymbol{X}^T = (\boldsymbol{U}\boldsymbol{\Sigma}\boldsymbol{V}^T)(\boldsymbol{V}\boldsymbol{\Sigma}\boldsymbol{U}^T) = \boldsymbol{U}\boldsymbol{\Sigma}^2\boldsymbol{U}^T \tag{12}$$

$$\boldsymbol{C}' = \boldsymbol{X}^T\boldsymbol{X} = (\boldsymbol{V}\boldsymbol{\Sigma}\boldsymbol{U}^T)(\boldsymbol{U}\boldsymbol{\Sigma}\boldsymbol{V}^T) = \boldsymbol{V}\boldsymbol{\Sigma}^2\boldsymbol{V}^T \tag{13}$$



It follows that the $C$ and $C'$ have common eigenvalues, each of which is equal to the squared singular value of $X$, and that $U$ and $V$ in Eqn. (11) correspond to the eigenvectors of $C$ and $C'$ , respectively.

Additionally, from the relational expression of the singular value decomposition, data vector $x_i$ for case $i$ is expressed as the linear combination of the eigenvectors of the covariance matrix as

$$x_i = \sum_{k=1}^{p} (\sqrt{\lambda_k} v_{ik}) u_k = \sum_{k=1}^{p} \alpha_{ik} u_k = \alpha_{i1} u_1 + \cdots + \alpha_{ip} u_p \tag{14}$$

where $v_{ij}$ are components of $V$ with $i$ and $j$ indicating row and column numbers, respectively. Here, the modes are arranged in accordance with the descending order of the corresponding eigenvalues and $\alpha_{ik}$ is the coefficient of the $k$-th mode associated

with the $i$-th case. Once modes to be omitted are determined according to the magnitude relationships of the contribution rates, etc., the number of modes $r$ used to approximate the data is determined, so that the equation above yields the following reduced order expression:

$$\hat{x}_i = \sum_{k=1}^{r} \alpha_{ik} u_k = \alpha_{i1} u_1 + \cdots + \alpha_{ir} u_r \tag{15}$$

where $\hat{x}_i$ is expected to be a close approximation of the original data $x_i$ for the $i$-th case. In this manner, data for a certain case

can be expressed as a linear combination of major modes expressing the fundamental features. Remember that each of these selected modes corresponds to the eigenvector of the covariance matrix whose associated eigenvalue has relatively larger then others. As acknowledged above, error may occur in Eqn. (15) due to the truncation or mode omission.

### 2.3    Construction of the surrogate model

Once the reduced order expressions of all the data vectors are obtained, a surrogate model can be obtained straightforwardly.

We begin with identifying the relationship between the coefficient for each mode and the set of input parameters $\beta$ , each of which corresponds to a selected scenario or case. That is, coefficients $\alpha_{ij}$ are approximated as functions of parameters $\beta$ by interpolation or regression, which are denoted by $f_j(\beta)$. Then, the surrogate model can be expressed in the following equation whose independent variables are input parameters $f_j(\beta)$:

$$\hat{x}(\beta) = \sum_{k=1}^{r} f_k(\beta) u_k \tag{16}$$

In this study, $f_j(\beta)$ is determined by the interpolation with radial basis functions (RBF) (Buhmann, 1990) because it is applicable even if the parameters are not distributed in equidistant intervals and even if some defects are present in the data.

RBF interpolation for $f_j(\beta)$ can be expressed by the following equation:

$$f(\beta) = \sum_{i=1}^{N} w_i \phi(\beta, \beta_i) = \exp(-\gamma ||\beta - \beta_i||^2) \tag{17}$$

where $\beta_i$ contains the set of parameters for case $i$, $w_i$ is the weight, and $\phi(\beta, \beta_i) \equiv \exp(-\gamma ||\beta - \beta_i||^2)$ are RBF. Here, $\gamma$ is the

parameter controlling the smoothness of the function. Also, by substituting the set of actual the input parameters in Eq. (17),





we obtain the following simultaneous equations to determine the set of weights $w_i$:

$$
\begin{pmatrix} f(\boldsymbol{\beta}_1) \\ \vdots \\ f(\boldsymbol{\beta}_N) \end{pmatrix} = \begin{pmatrix} \phi(\boldsymbol{\beta}_1, \boldsymbol{\beta}_1) & \cdots & \phi(\boldsymbol{\beta}_1, \boldsymbol{\beta}_N) \\ \vdots & & \vdots \\ \phi(\boldsymbol{\beta}_N, \boldsymbol{\beta}_1) & \cdots & \phi(\boldsymbol{\beta}_N, \boldsymbol{\beta}_N) \end{pmatrix} \begin{pmatrix} w_1 \\ \vdots \\ w_N \end{pmatrix}
\tag{18}
$$

In this study, ridge regression (Hoerl and Kennard, 1970) is used to compute weights for Eq. (18).By introducing the ridge regression, it is possible to prevent the overfitting problem. Since the accuracy of the interpolation depends on the regularization

parameter used in the ridge regression and the smoothness parameter $\gamma$ of the RBF interpolation, these values are determined by the application of a certain cross-validation technique (Stone, 1974) combined with the Bayesian optimization (Močkus, 1975) in this study.

## 3   Application to a real disaster case

In this section, the proposed method is applied to tsunamis that have occurred. The simulation and the construction of the

surrogate model are based on the tsunami induced by the 2011 earthquake that occurred off the Pacific coast of Tohoku.

### 3.1   Tsunami simulation

In this study, as previously mentioned, the calculation was performed in two steps, including a numerical analysis based on the 2011 earthquake that occurred off the Pacific coast of Tohoku using 2D analysis over a wide area and 3D runup analysis for the target region. To construct a surrogate model that considers uncertainty, it is first necessary to decide what uncertainty

should be considered. Next, we considered the dominant factors from the occurrence of the tsunami to its runup. For example, the epicenter position and magnitude are factors that need to be considered during the earthquake. During tsunami propagation, the fault model that controls the initial waveform of the tsunami and the submarine topography data need to be included. In this study, the 2011 earthquake that occurred off the Pacific coast of Tohoku was used as a basis. That is, the relevant factors in the process of the earthquake itself can be found in the accomplishments of research conducted up to this point. In concrete terms,

in our study we used a fault model (Fujii-Satake model Ver. 8.0) (Satake et al. (2013)) composed of the 55 small faults shown in Fig.2. By considering uncertainty in terms of the two parameters of slip and rake in Fig.3, which are thought to have a deep relationship with the characteristics of fault stagger, we conducted a numerical analysis of multiple scenarios using different values of these parameters. The cases that were analyzed and the names of the cases are shown in Table 2. For the slip of each small fault, there were five patterns from 0.7 to 1.4, and for the rake, there were 10 patterns with $-20°$ and $+25°$ changes in

the angle, making a total of 50 cases.

### 3.1.1   Wide-area 2D tsunami analysis

To obtain the information about the tsunami wave usable for the 3D analysis, a tsunami analysis is carried out utilizing the 2D shallow water flow model. First, a wide-area 2D analysis was performed to obtain time history data for the tsunami height


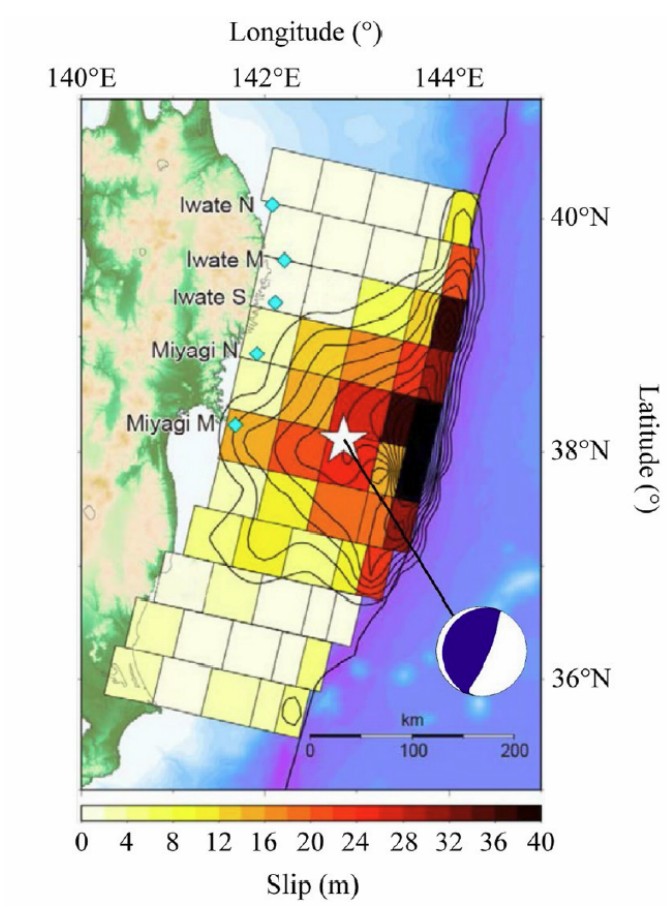

**Figure 2.** Fujii-Satake model Ver. 8.0 (adapted from Kotani et al. (2020)).

**Table 1.** Calculation cases

| | | | Rake | | | | | | | | | |
|---|---|---|---|---|---|---|---|---|---|---|---|---|
| | | [°] | -20 | -15 | -10 | -5 | 0 | +5 | +10 | +15 | +20 | +25 |
| | [%] | Normalized value | 0.753 | 0.815 | 0.877 | 0.938 | 1 | 1.062 | 1.123 | 1.185 | 1.247 | 1.309 |
| | 70 | 0.7 | S1R1 | S1R2 | S1R3 | S1R4 | S1R5 | S1R6 | S1R7 | S1R8 | S1R9 | S1R10 |
| | 85 | 0.85 | S2R1 | S2R2 | S2R3 | S2R4 | S2R5 | S2R6 | S2R7 | S2R8 | S2R9 | S2R10 |
| Slip | 100 | 1 | S3R1 | S3R2 | S3R3 | S3R4 | S3R5 | S3R6 | S3R7 | S3R8 | S3R9 | S3R10 |
| | 120 | 1.2 | S4R1 | S4R2 | S4R3 | S4R4 | S4R5 | S4R6 | S4R7 | S4R8 | S4R9 | S4R10 |
| | 140 | 1.4 | S5R1 | S5R2 | S5R3 | S5R4 | S5R5 | S5R6 | S5R7 | S5R8 | S5R9 | S5R10 |




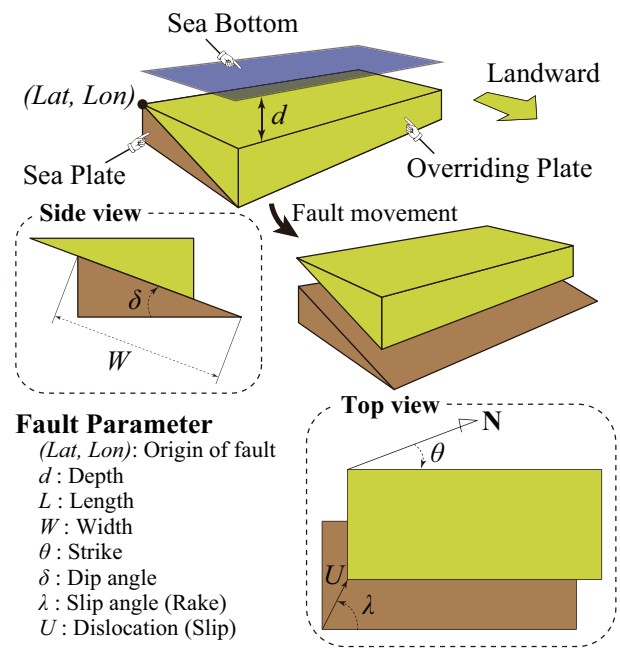

**Figure 3.** Illustration of the fault parameters (adapted from Kotani et al. (2020)).

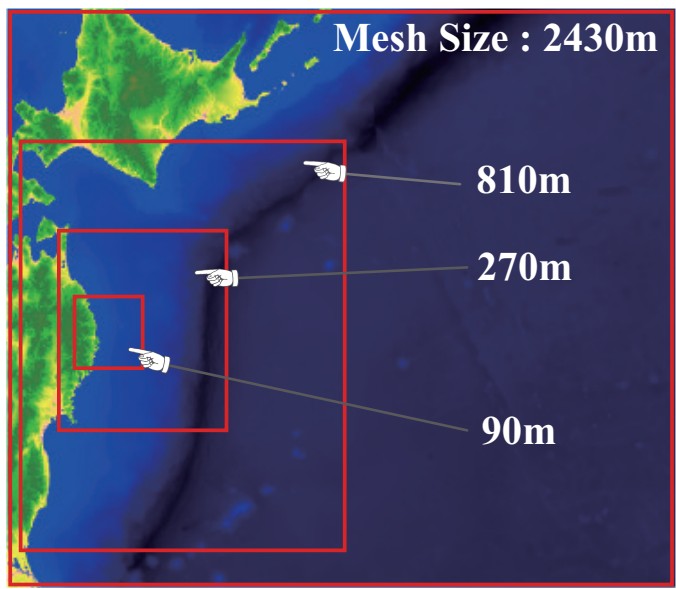

**Figure 4.** Nested analysis regions (adapted from Kotani et al. (2020)).

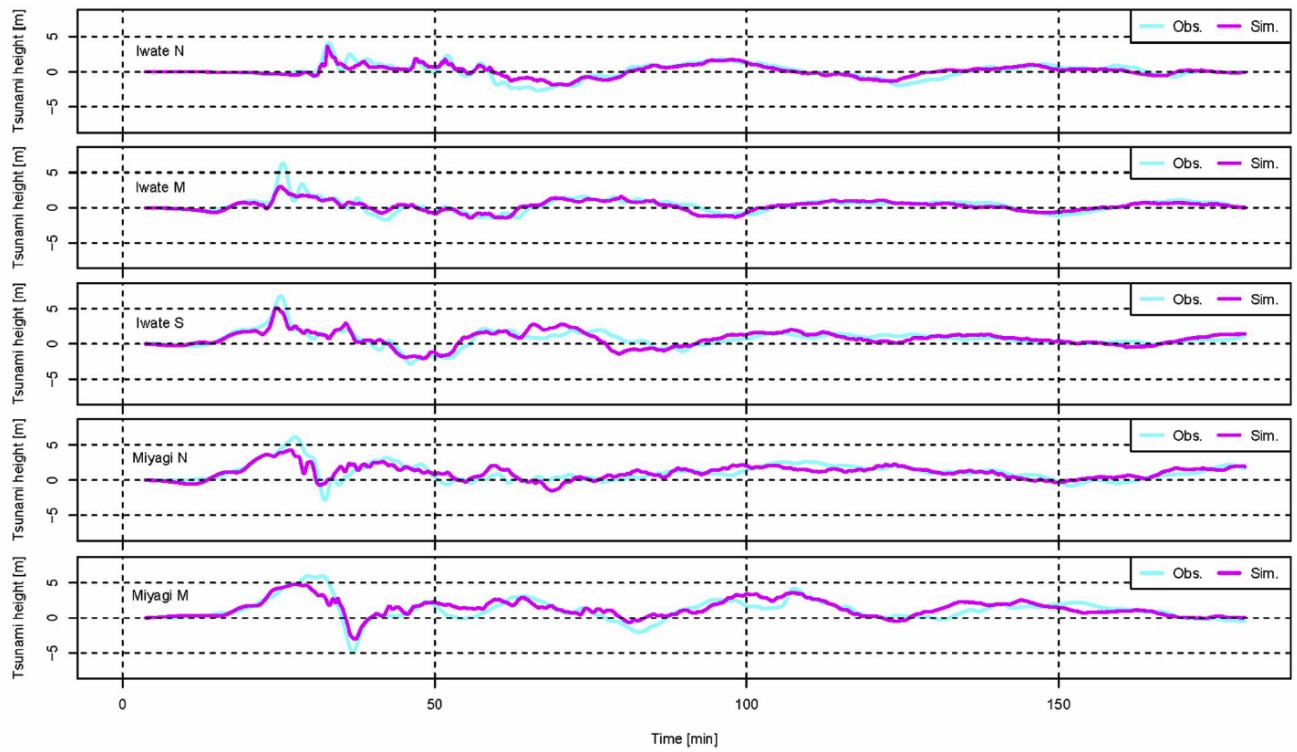

**Figure 5.** Comparison between observational data and simulation results (adapted from Kotani et al. (2020)).

and flow velocity observed within the bay of the target region. Regarding the initial waveform of the tsunami, Okada (1992)

was employed. The fluctuation vertical component close to the seabed was calculated based on the fault model and the initial

water level fluctuation was determined. Additionally, an analysis mesh was created with intervals of 2430 [m], 810 [m], 270

[m], and 90 [m], and nesting was performed to subdivide this mesh in relation to the target based on the tsunami wave source.

An image of this nesting is shown in Fig. 4. To confirm the reasonableness of the obtained 2D analysis, we compared the

results to the actual measured values of the fluctuation in the tsunami water level in the 2011 earthquake that occurred off the

Pacific coast of Tohoku. As a point of comparison, we used the GPS wave meter observation data of the Nationwide Ocean

Wave information network for Ports and Harbours  (NOWPHAS) provided by the Port Bureau, Ministry of Land Infrastructure

Transport and Tourism. The observation positions were offshore to the north of Iwate Prefecture (No. 1), offshore to the center

of Iwate Prefecture (No. 2), offshore to the south of Iwate Prefecture (No. 3), offshore to the north of Miyagi Prefecture (No.

4) and offshore to the center of Miyagi Prefecture (No. 5), and a comparison of these is shown in Fig. 5. The result is that the

actual measured values were sufficiently reproduced, and a 3D tsunami analysis was performed using these results as input

values.



### 3.1.2   Connecting the 2D and 3D analyses

Using the results of the 2D tsunami analysis over a wide area as the input conditions, a 3D tsunami analysis was performed to represent the tsunami runup in the target region. In terms of the concrete connection method, the method used in the study of

Takase et al. (2016) was employed. An image of the location of 2D-3D boundary is shown in Fig. 6. The time-series data of the wave height and flow velocity obtained from the 2D wide-area analysis were used as 3D input values.

### 3.1.3   3D runup analyses

Tsunami runup was represented by the 3D analysis using SFEM on the 2145 [m]×2600 [m] field shown in Fig. 7. Images of the FE mesh used in this analysis are described in Fig. 8. A fine mesh with a minimum mesh size of 1.5m is used around

buildings and a coarse mesh with maximum mesh size of 10 m is used for mountainous areas. The non-slip boundary condition is used on building surfaces and ground surface. Fig. 9 shows snapshots of the tsunami runup obtained in the 3D simulation. A very complex 3D flow that cannot be represented in 2D simulations is seen in the simulated results. As mentioned in subsection 4.1, the calculation condition of Case S3R5 corresponds to the real tsunami induced by the 2011 earthquake that occurred off the Pacific coast of Tohoku. Therefore, we checked the accuracy of the simulated result obtained in Case S3R5 by comparing

with survey results. Data of the maximum inundation heights are provided by the 2011 Tohoku Earthquake Tsunami Joint Survey Group (2012). Fig. 10 indicates a comparison between simulated maximum inundation heights and survey data. The evaluation points are shown in Fig. 11. The comparison shows that the simulated inundation heights near the coastline (points A, B, C, and E) well represent the real inundation heights, while there are big gap at other points. The reason for this big gap is that buildings are assumed to be rigid bodies and the buildings are not washed away in the 3D analysis. Although most of

buildings were washed away due to the tsunami, buildings suppress tsunami runup in the analysis. While this suggests it is better to consider the effect of washing away buildings, modeling this effect is difficult. Furthermore, the main objective of this study is to construct a surrogate model of tsunami force and discuss it's performance. We therefore conclude that the simulated results roughly express the real tsunami, that use the simulated results in this study.

In order to construct a surrogate model of tsunami force, we can use the hydrodynamic force acting on buildings calculated

in the 3D simulation. It is however difficult to quantify pointwise tsunami force because the force is strongly affected by the direction of building surfaces. To avoid this problem, we consider a 2D mesh with a grid size of 10m for evaluating the tsunami force. An image of the mesh is shown in Fig. 12. The force acting on the building surface is integrated in each mesh and quantified as a scalar value. The spatial distribution of the tsunami force was obtained in this manner.

### 3.2   Application of proper orthogonal decomposition

### 3.2.1   Definition of the target risk indicators and data matrix

In this study, the tsunami force acting on the buildings and the inundation depth are used as the target risk indicators, and the values integrated or averaged in the 10m mesh are used as the mean values. As the field is 2145 [m]×2600[m], the number of





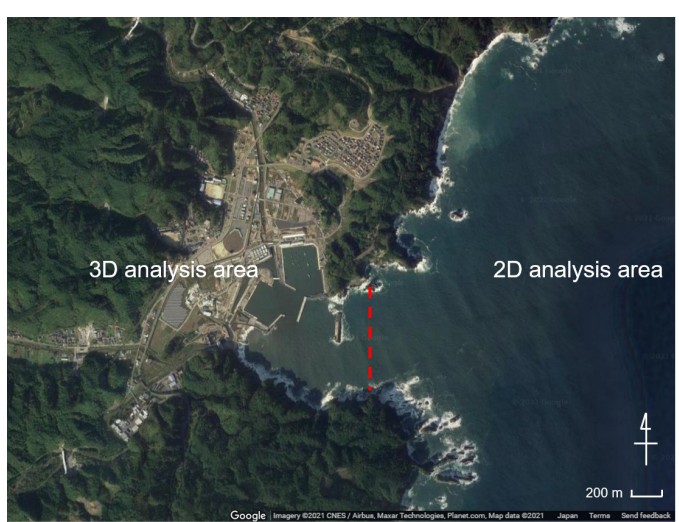

**Figure 6.** Boundary between the 2D analysis and 3D analysis areas. (© Google Maps)

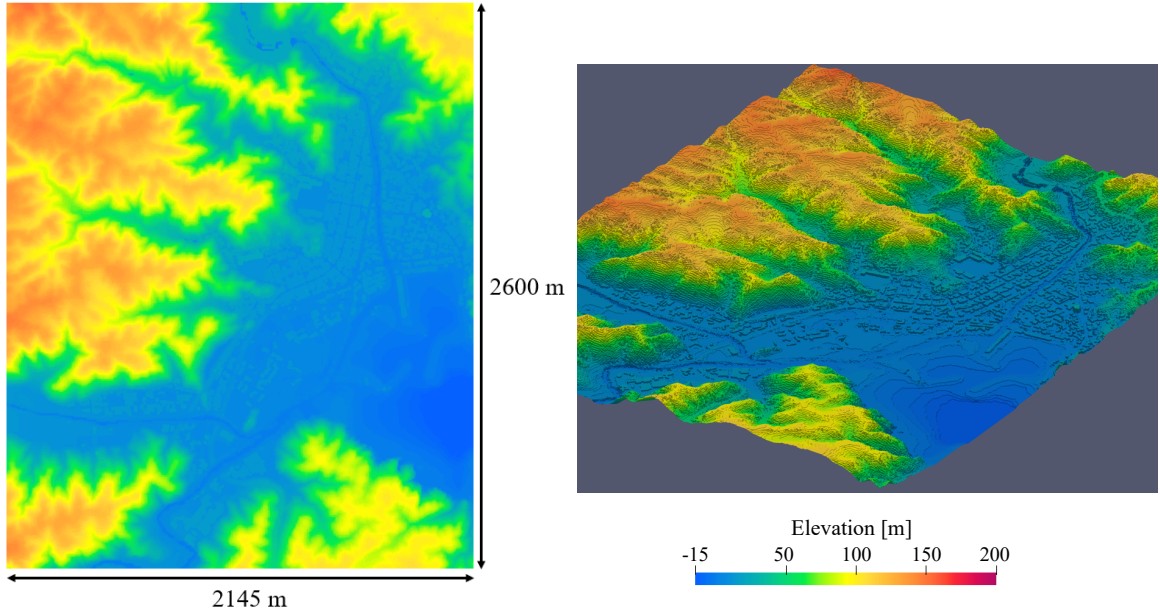

**Figure 7.** Analysis area.


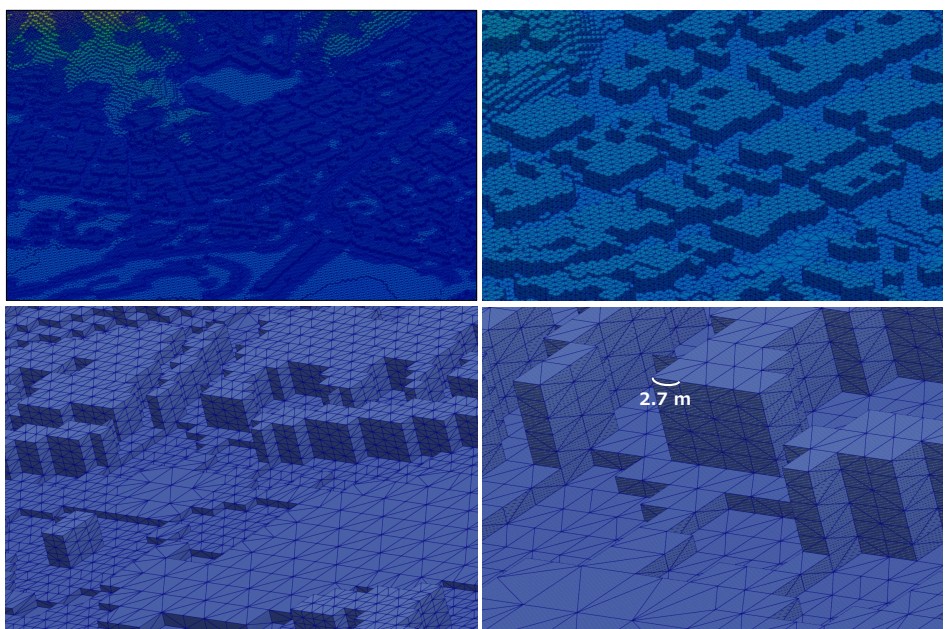

**Figure 8.** Images of FE meshes at different scales.

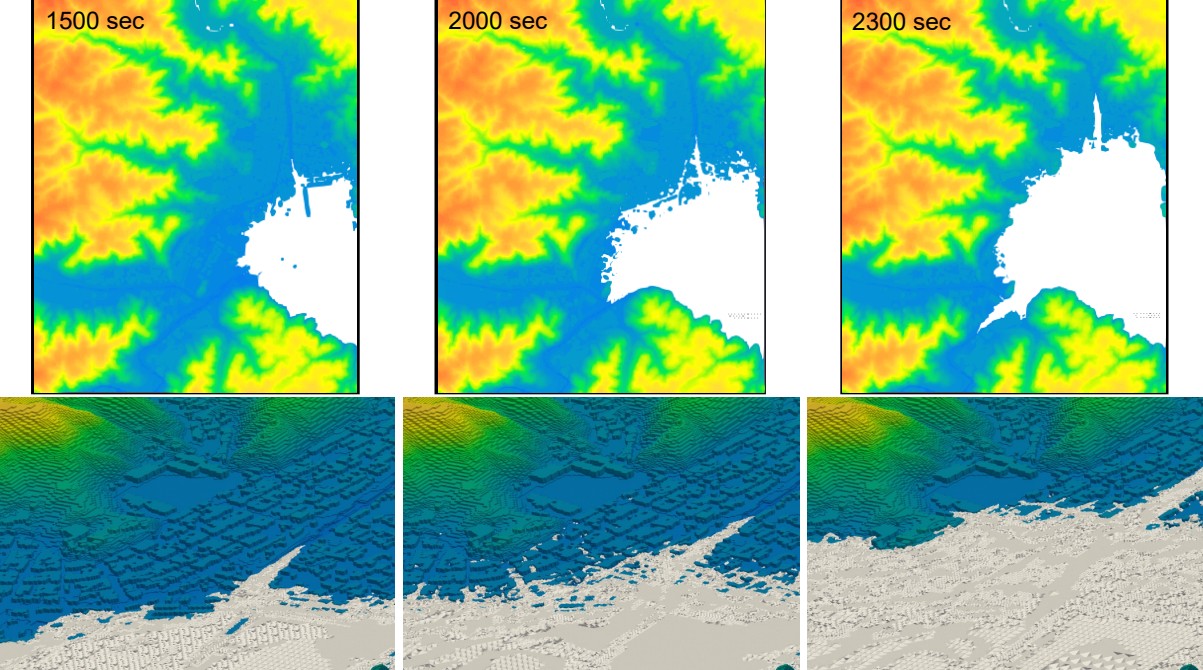

**Figure 9.** Snapshots of tsunami runup obtained in 3D analysis.




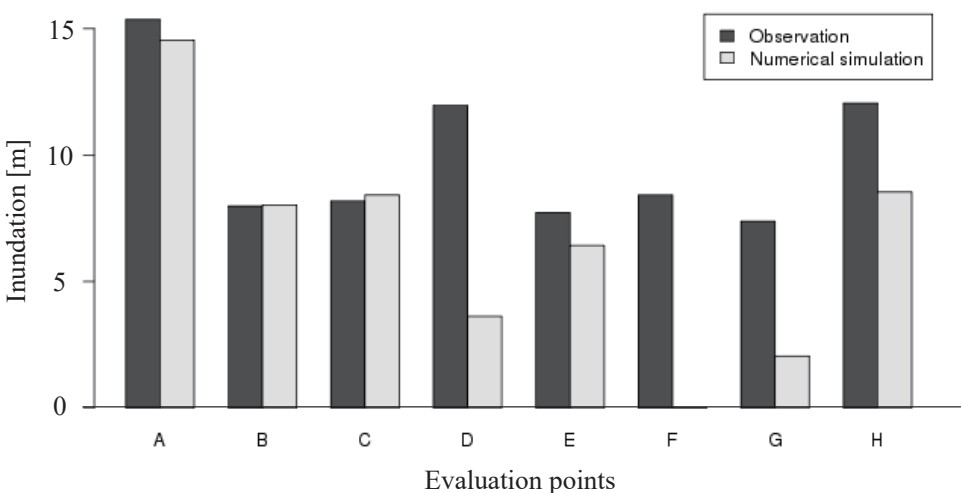

**Figure 10.** Comparison of inundation height between observation and numerical simulation.

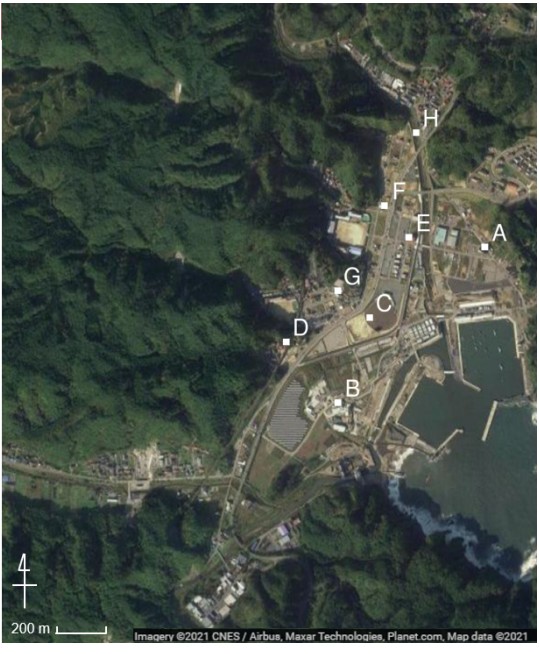

**Figure 11.** Points of comparison of inundation depth. (© Google Maps)




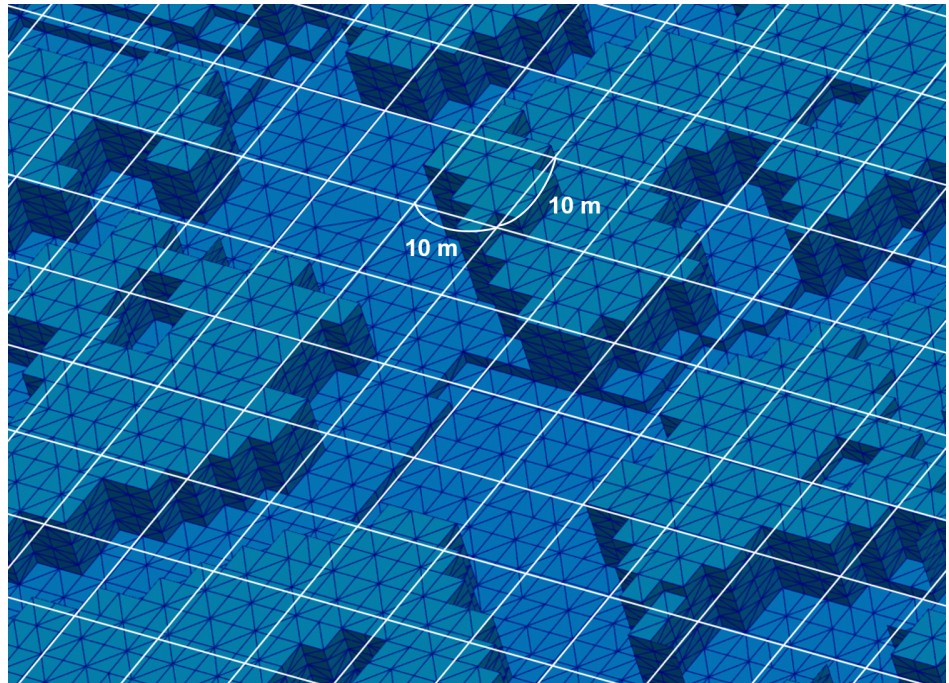

**Figure 12.** An Image of mesh for evaluating tsunami force.

evaluated points is $n = 214 \times 260 = 55640$. The impact force is calculated by using the pressure and the surface affected by the pressure in a 10-meter square. Additionally, 20 seconds is used for the time interval for the use data and 150 steps of data are

used in accordance with the time for the tsunami runup. In this application example, in addition to the two input parameters, time is introduced as another parameter because when predicting damage in real-time, it is necessary to evaluate the state of time-related developments in addition to spatial distribution.

Proper orthogonal decomposition is applied to construct a surrogate model for the results of 40 of the 50 cases shown in Table 1, excluding the 10 cases where the rake is $\pm 10°$. The reason that these 10 cases are not used for surrogate model construction

is that their associated data are later used to verify the reasonableness of the created surrogate model. Additionally, as 40 cases secured a sufficient number of analysis cases for the two variables in this study, verification is not performed, but it is desirable to assess how many cases are required when constructing the surrogate model.

Next, the data matrix is defined. First, the time-series data for a particular scenario can be defined as follows as matrix $\boldsymbol{X}_i$.

$$
\boldsymbol{X}_i = \begin{pmatrix} | & & | \\ \boldsymbol{x}(U_i, \lambda_i, t_1) & \cdots & \boldsymbol{x}(U_i, \lambda_i, t_m) \\ | & & | \end{pmatrix} \tag{19}
$$

Here, $\boldsymbol{x}(U, \lambda, t)$ is a vector showing the spatial distribution of the risk indicator in relation to time. Additionally, as a concrete parameter, $U_i$ expresses slip in relation to scenario (slip), and $\lambda_i$ expresses rake in relation to scenario $i$. Additionally, $m$



expresses the maximum number of output steps in the numerical analysis, and in the example in this study, this is $m = 150$. Data matrix $\boldsymbol{X}$ is defined as follows, as laid out in the column direction.

$$\boldsymbol{X} = (\boldsymbol{X}_1 \cdots \boldsymbol{X}_N) \tag{20}$$

The previously described proper orthogonal decomposition was carried out for this data matrix. By applying proper orthogonal decomposition to this matrix, it is possible to extract a common temporal mode for data including time, and it is possible to construct a surrogate model including the time direction.

### 3.2.2   Extracted spatial modes and construction of a surrogate model

Here, we show the results of applying proper orthogonal decomposition in relation to impact force and inundation data.

First, in relation to the extracted spatial mode, the impact force and water depth for the first mode to the third mode are as shown in Figs. 13 and 14, respectively.

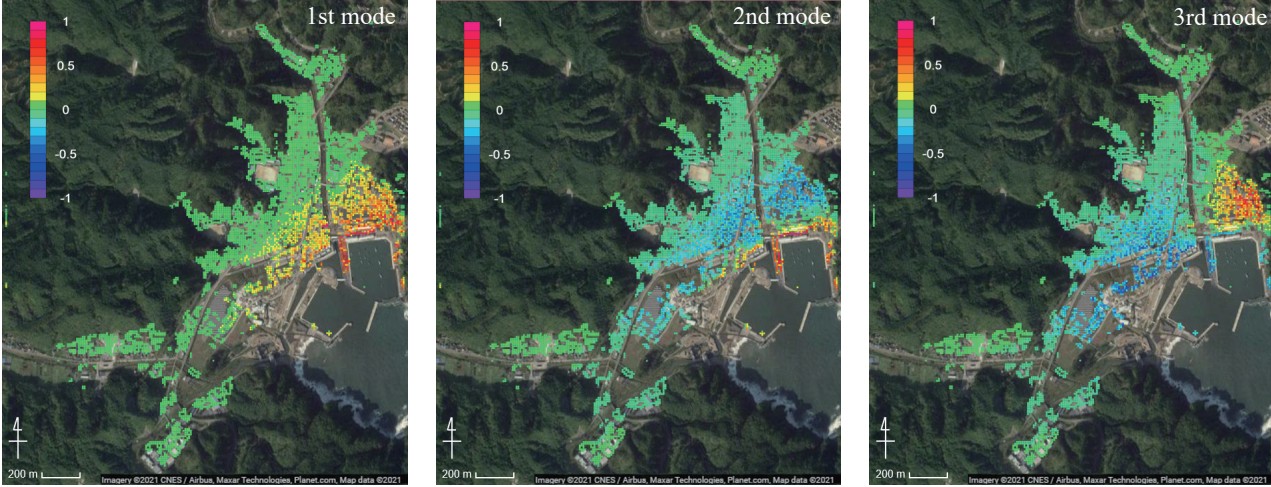

**Figure 13.** Spatial modes of impact force extracted by POD. (© Google Maps)

From these figures, it can be confirmed that the impact force distribution and inundation depth distribution both have roughly the same spatial distribution characteristics. If we examine the characteristics in more detail, in the case of the first mode, we can see the tendency for the coastal side to be most strongly impacted. Naturally, since the points close to the coast are most

affected by the tsunami, this is a mode that best expresses this kind of impact. Additionally, the second mode expresses opposite tendencies for the section close to the coast and the section behind it, and the third mode expresses opposite tendencies for the eastern and western coastal areas.

The contribution rates of the impact force and water depth are shown in Fig. 15. The contribution in the first mode is extremely high for both impact force and inundation depth. In this study, as an indicator of the number of modes, the error and




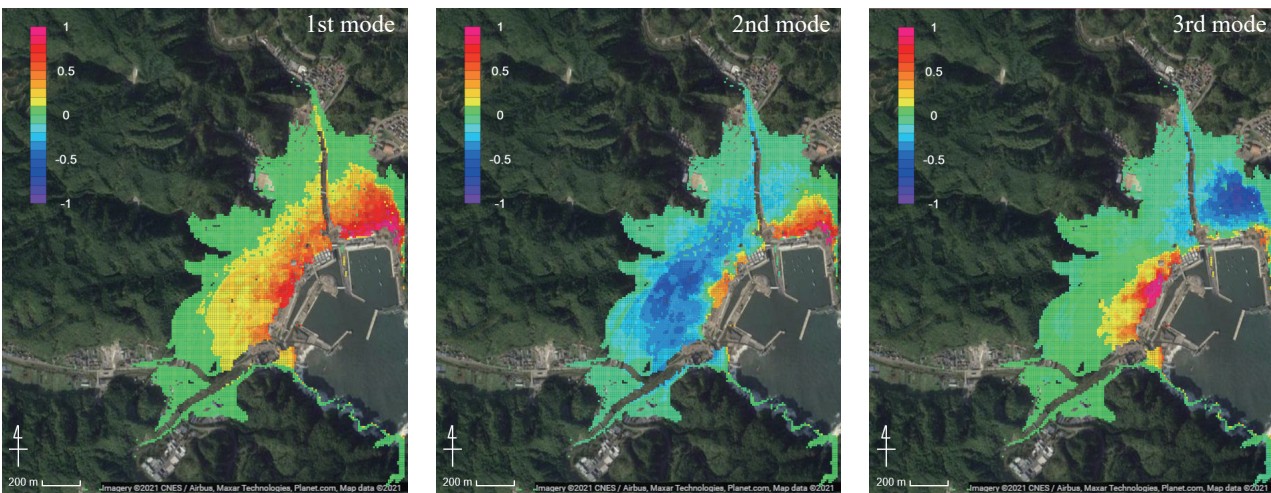

**Figure 14.** Spatial modes of inundation depth extracted by POD. (© Google Maps)

the contribution rate were investigated. In this example, for the 10 cases of data kept aside as verification data, the root mean squared error for the results obtained from the numerical analysis and the results obtained from the surrogate model for each mode are calculated. The error of case $e_i$ is calculated using the following equation.

$$e_i = \sqrt{\frac{1}{nm}||\boldsymbol{X}_i - \hat{\boldsymbol{X}}_i||_F^2} \tag{21}$$

Here, $n$ is the number of evaluation points and $m$ is the number of time step. Additionally, $\boldsymbol{X}_j$ is time series data of numerical 275 simulation results for $i$-th case and $\hat{\boldsymbol{X}}_j$ is results obtained from a surrogate model with $r$ modes. the relations between the number of modes and the error for each risk index is shown in Fig. 16. From Fig. 16, it can be seen that the error decreases as the number of modes used in the surrogate model increases, and as the number of modes increases, the error rates decrease, thus gradually converging. In this study, 20 modes are used to construct the surrogate model, because the error has been sufficiently reduced.

Next, by interpolating the proper orthogonal decomposition coefficient as a parameter function, the surrogate model is created. As shown in Eq. 20, as the data matrix contains the slip $U$, rake $\lambda$, and time $t$, which are parameters for numerical analysis in the column direction, the proper orthogonal decomposition coefficient is also expressed as a function of slip, rake and time. In concrete terms, this coefficient can be expressed as $f_k(U, \lambda, t)$ as in the following equation.

$$\hat{\boldsymbol{x}}(U, \lambda, t) = \sum_{k=1}^{r} f_k(U, \lambda, t)\boldsymbol{u}_k \tag{22}$$

**3.2.3 Reconstructing the numerical simulation results**

Here, we compare the results obtained from the numerical analysis and the results from reconstructing the original data using the constructed surrogate model. First, when comparing the time-series data, several evaluation points are set. In this study, the




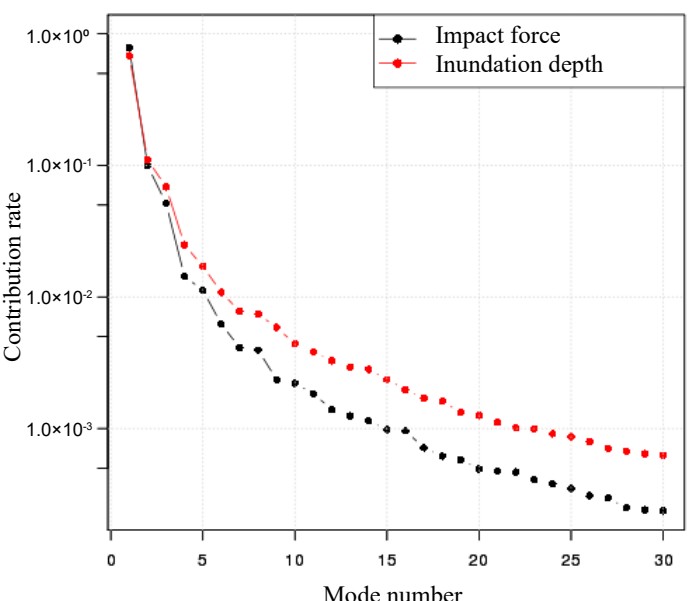

**Figure 15.** Contribution rate for each risk index

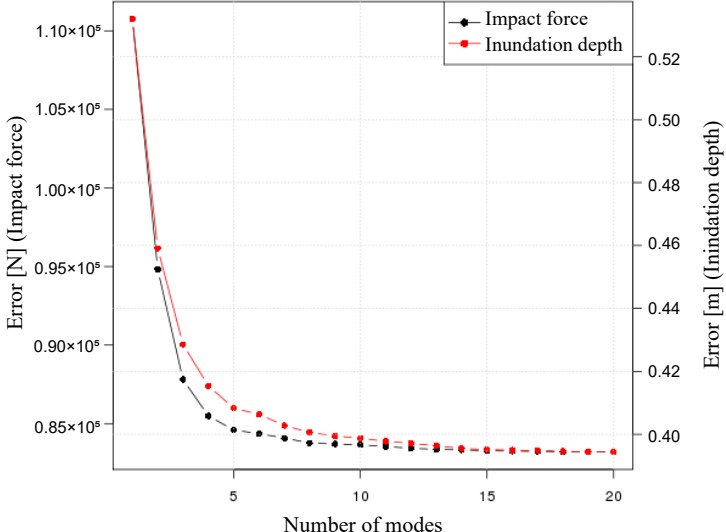

**Figure 16.** Relation error





10 points shown in Fig. 7 are set as evaluation points. Next, for the average case scenario S3R5, a comparison of the numerical analysis results and the results when reconstructing data using the surrogate model are shown. Snapshots of the impact force and water depth are shown in Figs. 18 and 19, and the time-series data for each physical quantity at points 2, 3 and 8 in Fig. 7 are shown in Fig. 20. Although only the time-series data obtained at the three points are shown in the figure, the same tendency is confirmed at other points. The black line indicates the results obtained from the actual numerical analysis, and the red line shows the results from reconstructing data using the surrogate model. Using the time-series data of the impact forces shown in Fig. 20 (a), it is possible to calculate the impulse acting on the buildings. The impulse can be obtained by integrating the force in time. Values of the impulse calculated by the numerical simulation and calculated by the surrogate model for the 10 evaluation points are shown in Fig. 21.

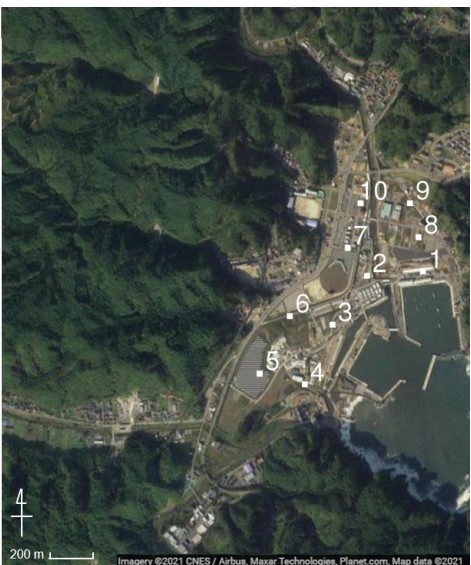

**Figure 17.** Evaluation points (© Google Maps)


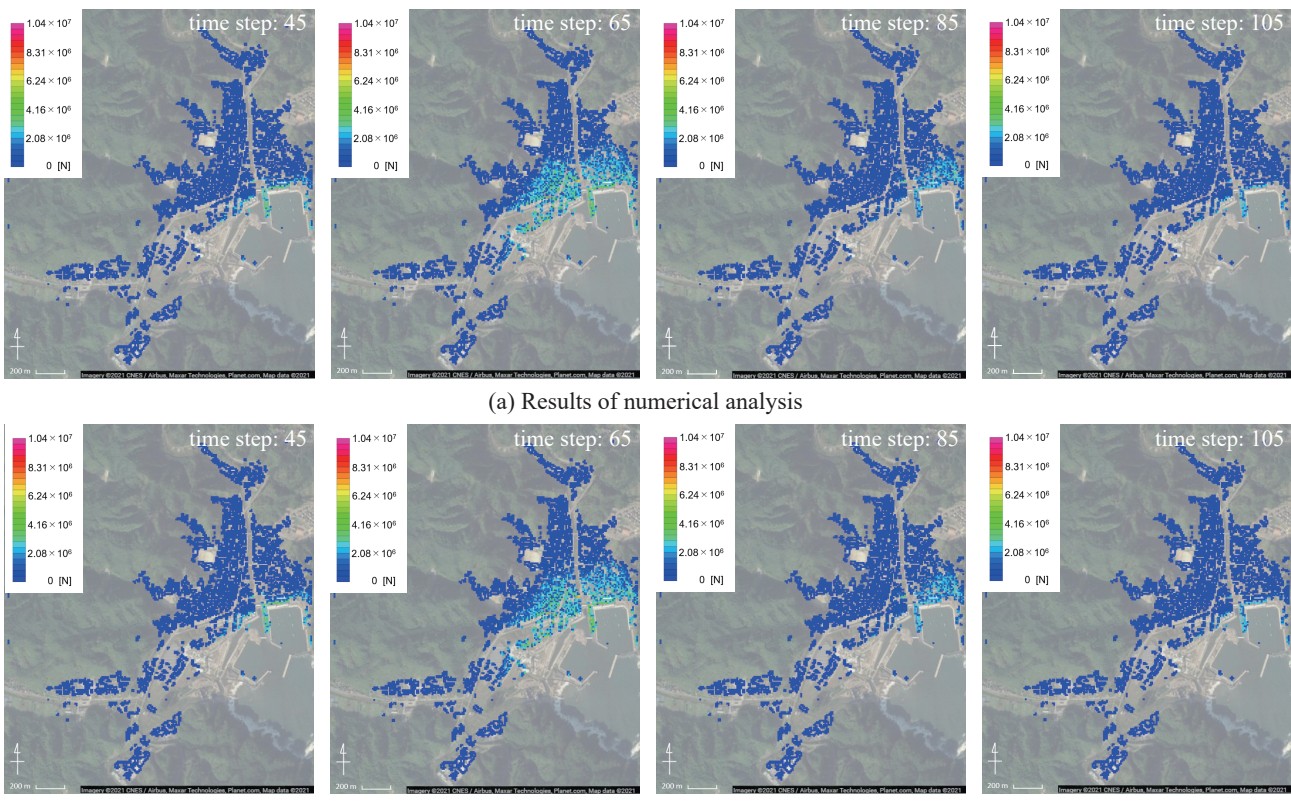

(a) Results of numerical analysis

(b) Results reconstructed by using spatial modes

**Figure 18.** Snapshots of the results obtained from numerical analysis and the results reconstructed by using spatial modes (impact forces) (© Google Maps)


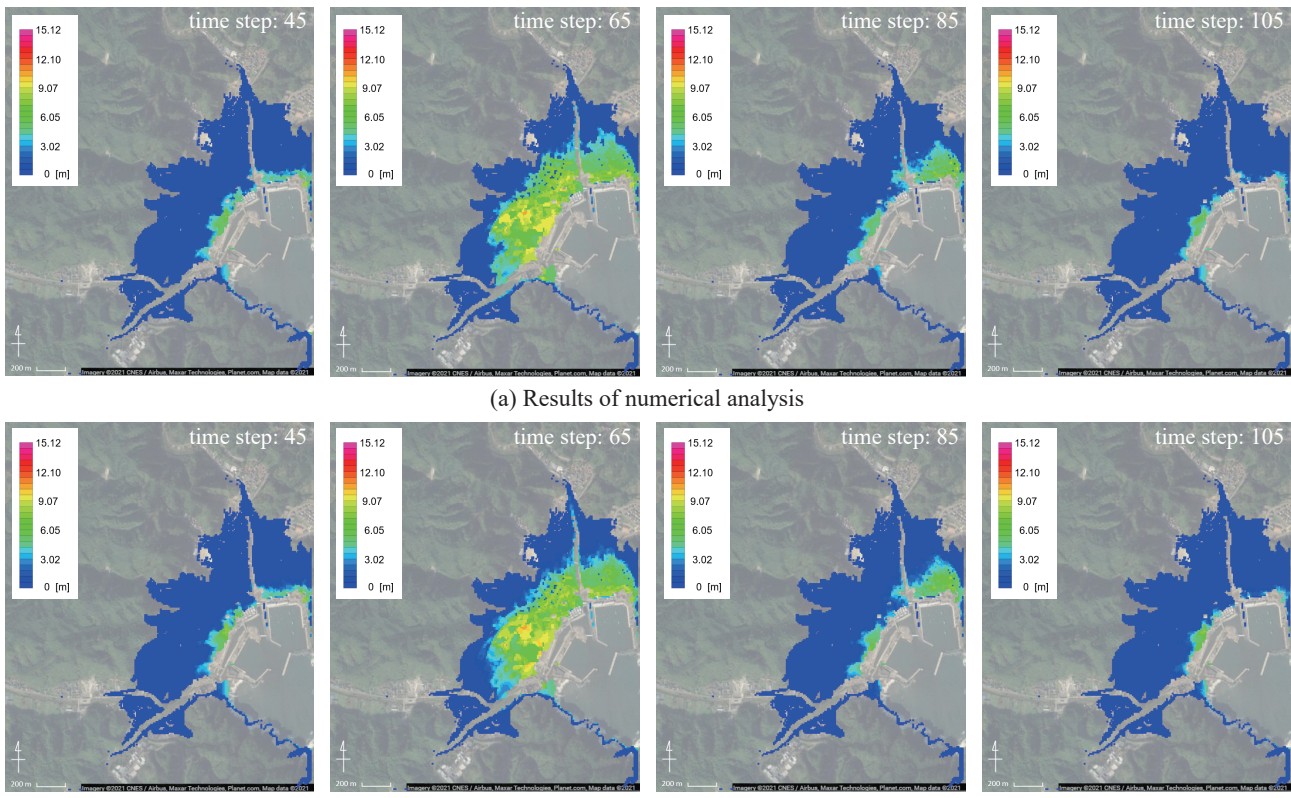

(a) Results of numerical analysis

(b) Results reconstructed by using spatial modes

**Figure 19.** Snapshots of the results obtained from numerical analysis and the results reconstructed by using spatial modes (inundation depth)
(© Google Maps)




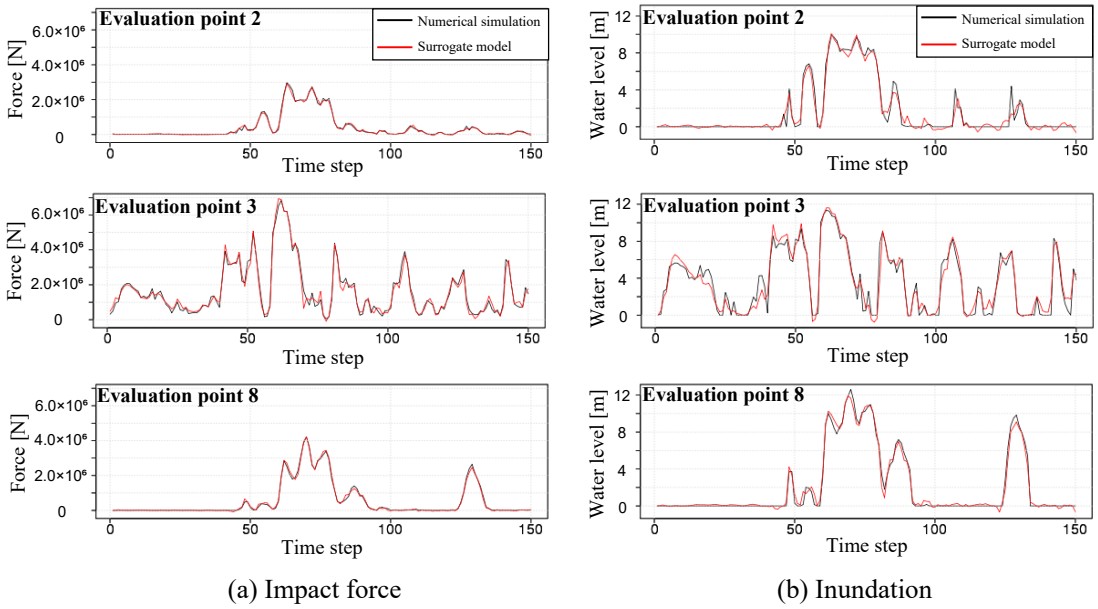

(a) Impact force                                (b) Inundation

**Figure 20.** Comparison of time-series data obtained from numerical analysis and the results reconstructed by using spatial modes

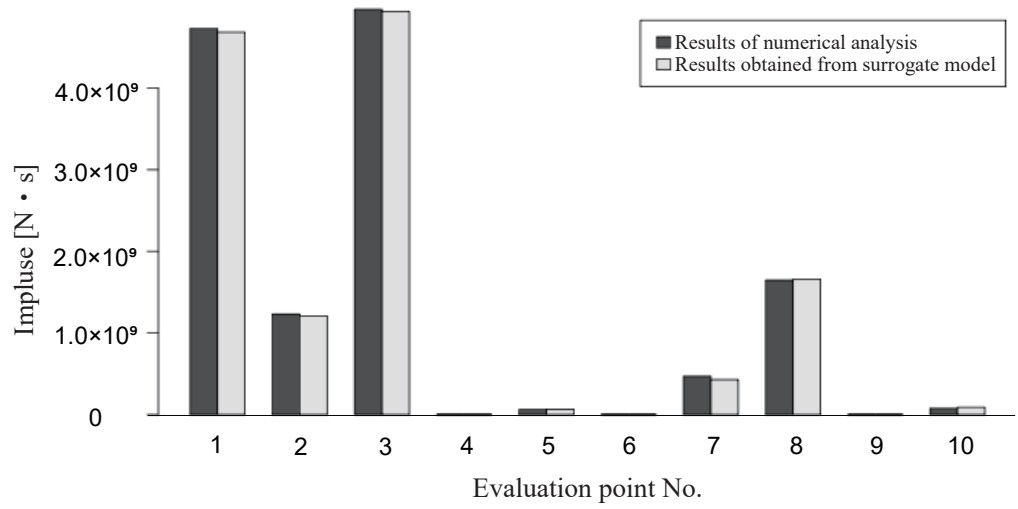

**Figure 21.** Comparison of impulses calculated from the results of numerical analysis and the results reconstructed by using spatial modes


Based on these results, it can be summarised that whereas errors partially occur based on the impact of mode reduction, in general, the original data are being reproduced, and it is possible to express the data as a linear combination of modes. It should also be noted that there are different tendencies between spatial distributions of tsunami force and those of inundation depth. For instance, at time step 65, high values are locally seen in the inundation depth, while that tendency is not seen in the result of tsunami force. This indicates that it's difficult to predict damage of buildings based on only inundation depth, and information of tsunami force is also respired for an accurate damage prediction.

### 3.2.4 Validation of the surrogate models

By comparing data that were not used for the construction of the surrogate model, it is possible to investigate whether the surrogate model could reproduce the numerical analysis results. A comparison of the results obtained from the numerical analysis and the surrogate model for the two scenarios of S2R3 and S5R7 is shown in Figs. 22-27. As in the previous section, for each physical quantity, a comparison by snapshots and time-series data at evaluation points 2, 3, and 8 is shown.

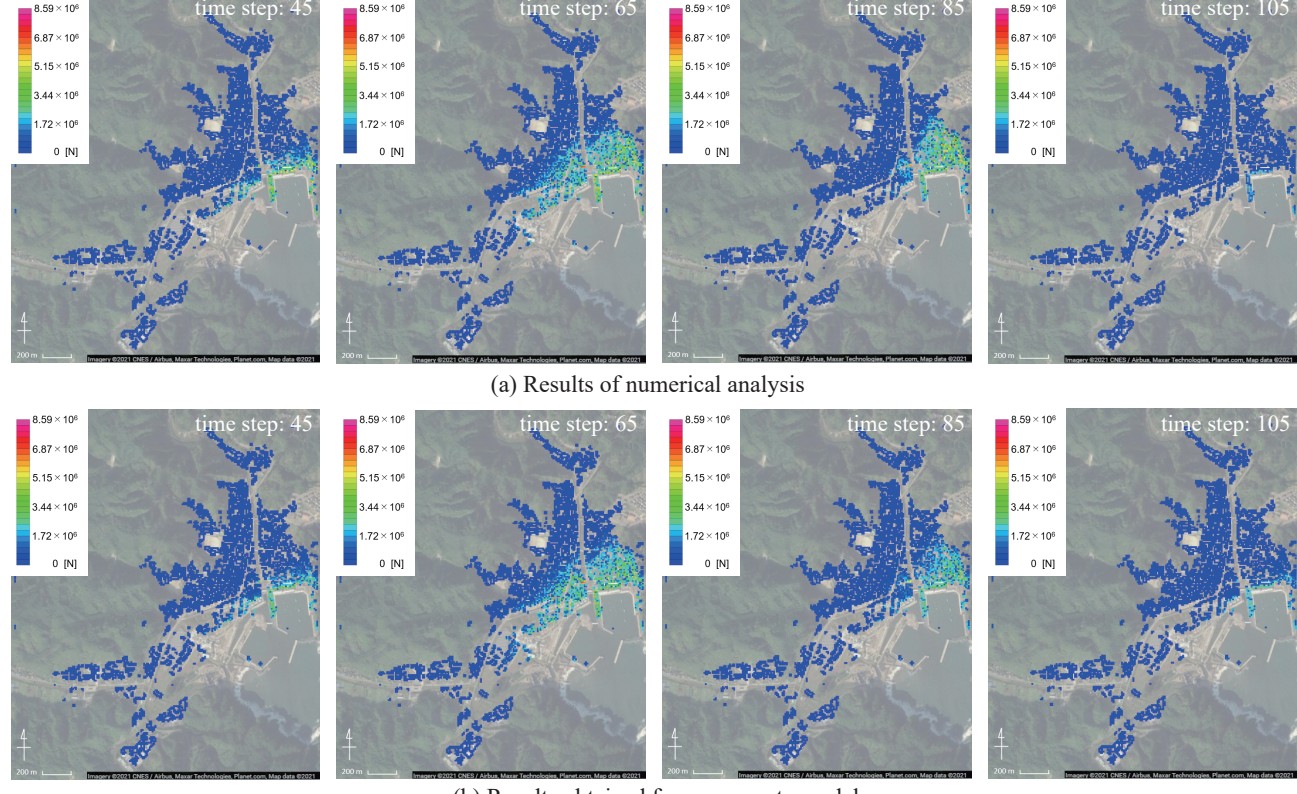

(a) Results of numerical analysis

(b) Results obtained from surrogate model

**Figure 22.** Snapshots of the results obtained from numerical analysis and results obtained from the surrogate model (S2R3, impact forces) (© Google Maps)

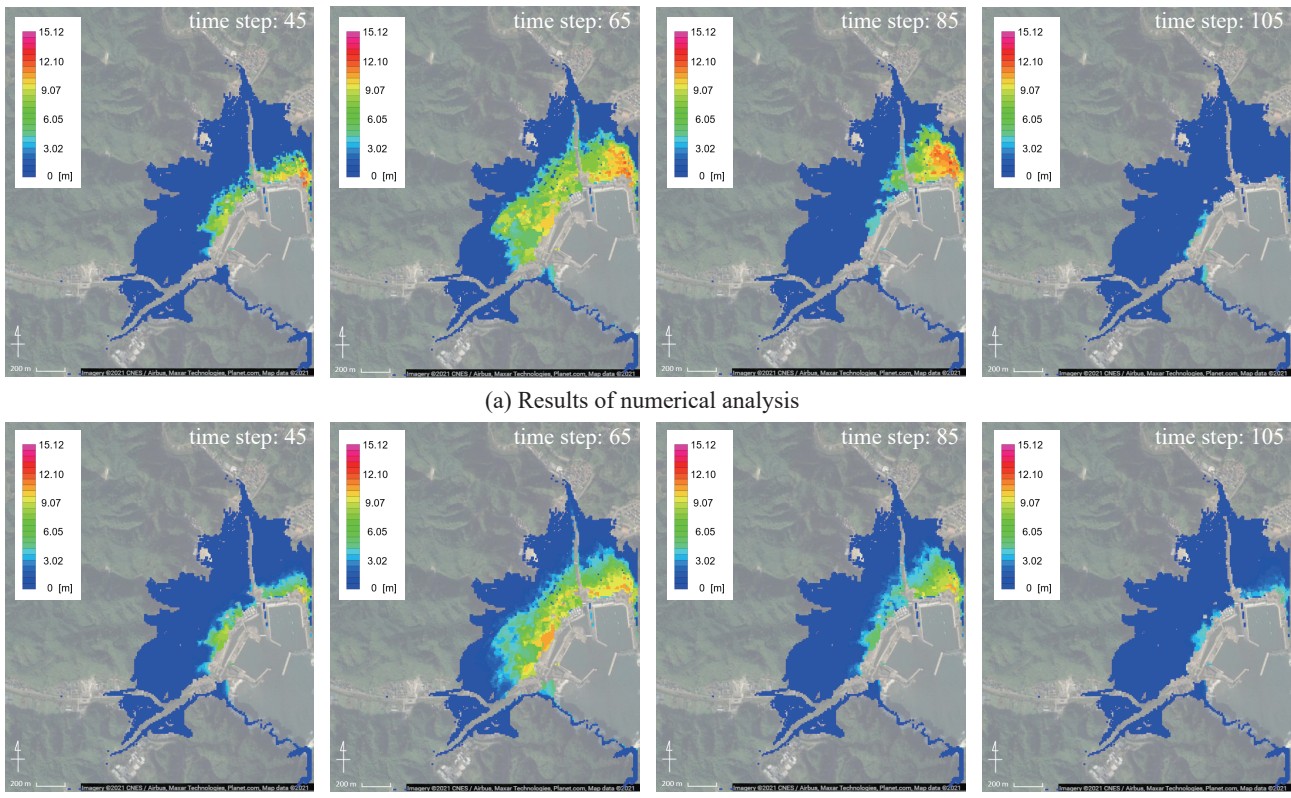

(a) Results of numerical analysis

(b) Results obtained from surrogate model

**Figure 23.** Snapshots of the results obtained from numerical analysis and the results obtained from the surrogate model (S2R3, inundation) (© Google Maps)

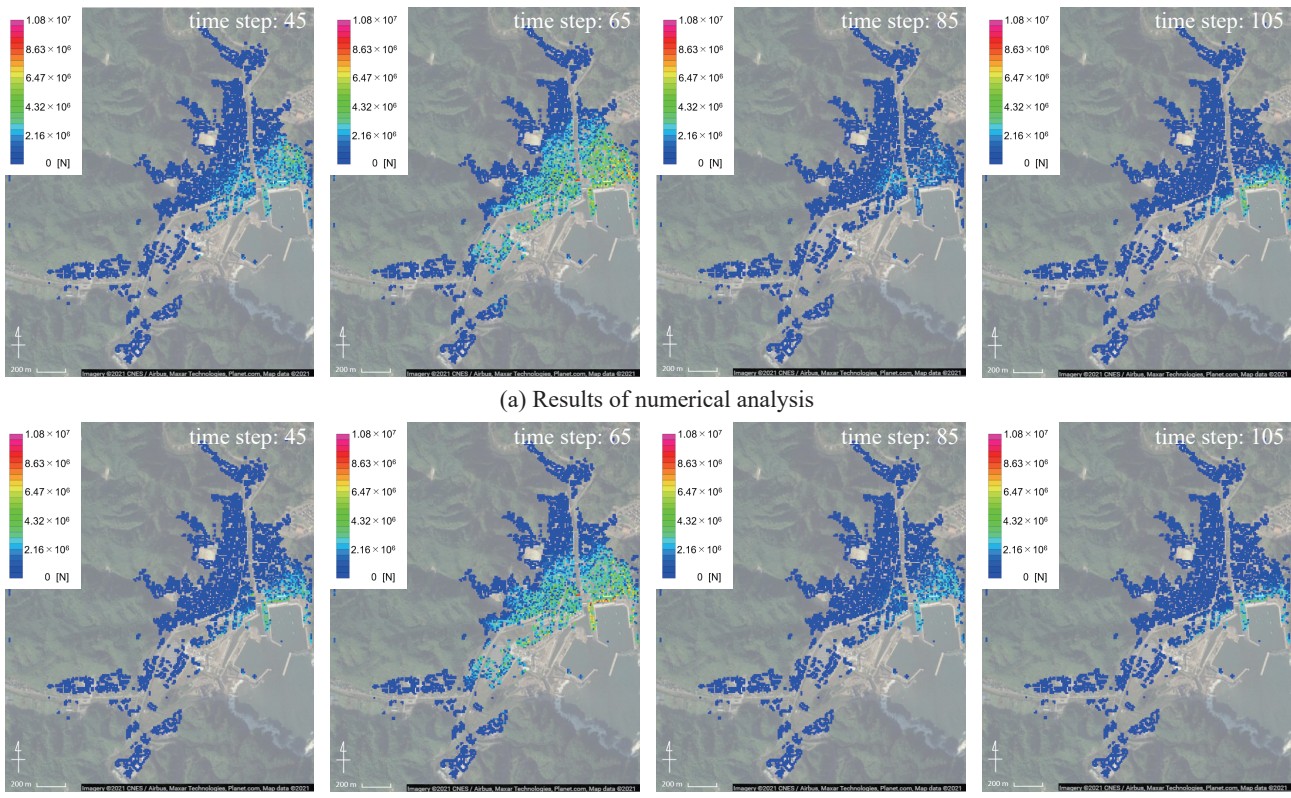

(a) Results of numerical analysis

(b) Results obtained from surrogate model

**Figure 24.** Snapshots of the results obtained from numerical analysis and results obtained from the surrogate model (S5R7, impact forces) (© Google Maps)


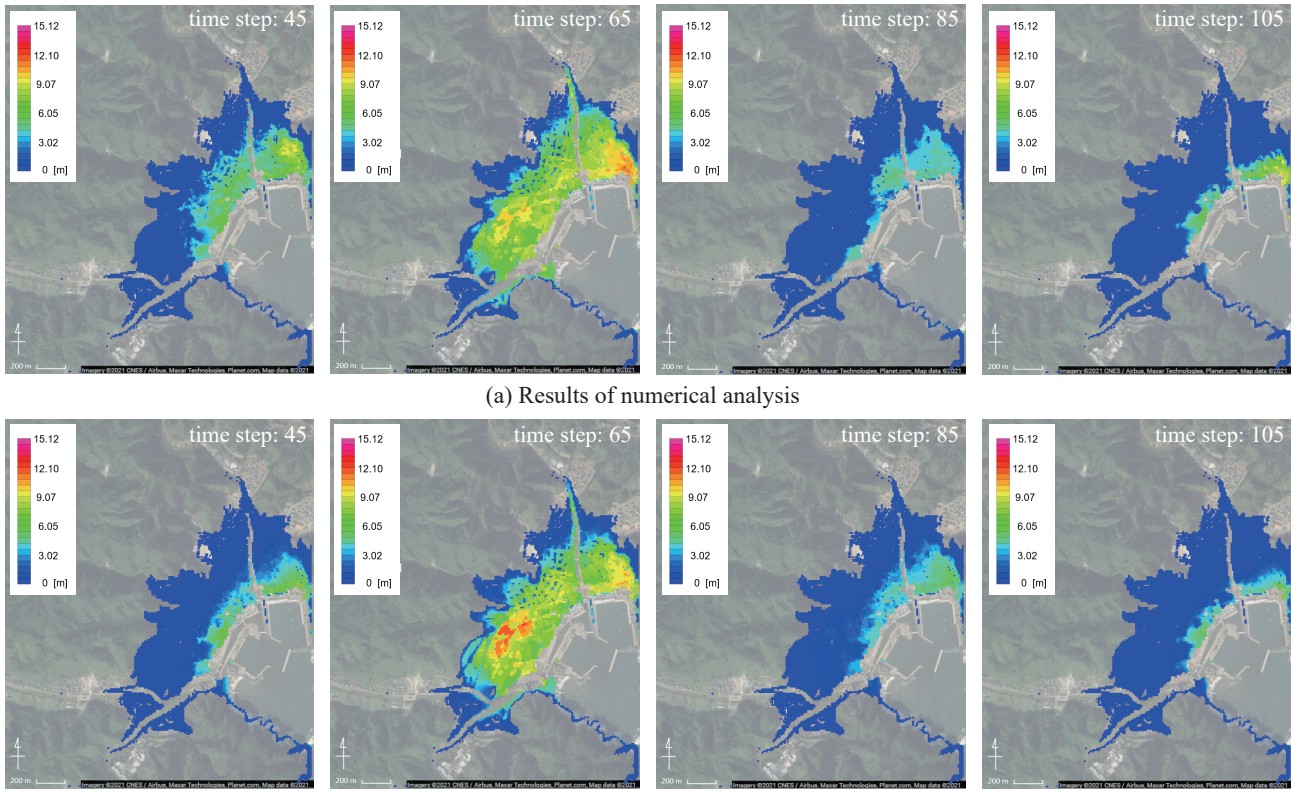

(a) Results of numerical analysis

(b) Results obtained from surrogate model

**Figure 25.** Snapshots of the results obtained from numerical analysis and the results obtained from the surrogate model (S5R7, inundation) (© Google Maps)



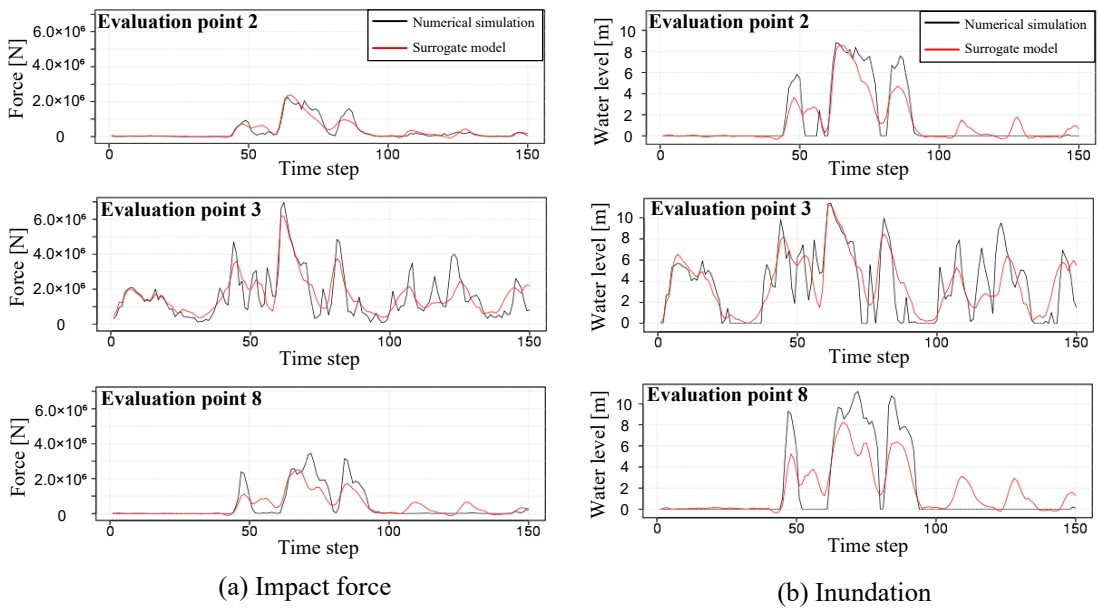

(a) Impact force        (b) Inundation

**Figure 26.** Comparison of time-series data obtained from numerical analysis and the surrogate model (S2R3)

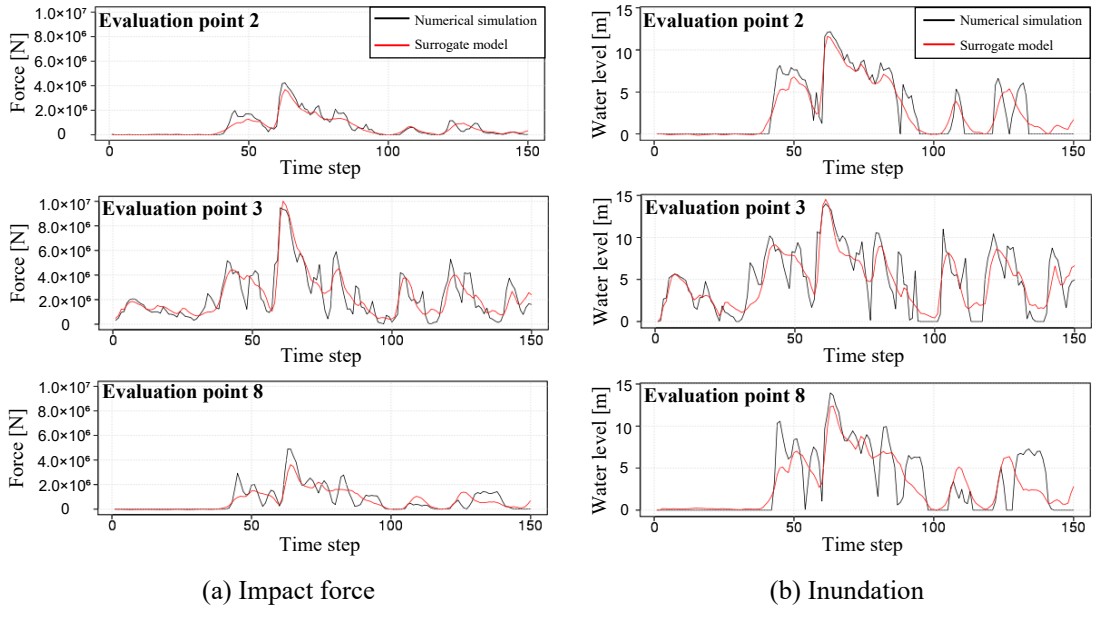

(a) Impact force        (b) Inundation

**Figure 27.** Comparison of time-series data obtained from numerical analysis and the surrogate model (S5R7)




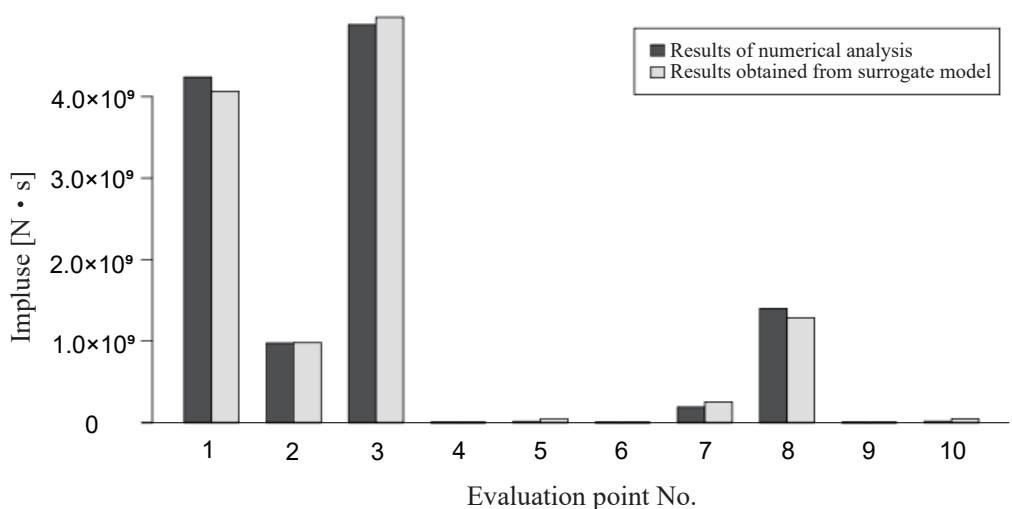

**Figure 28.** Comparison of impulses calculated from the results of numerical analysis and the results obtained from the surrogate model (S2R3)

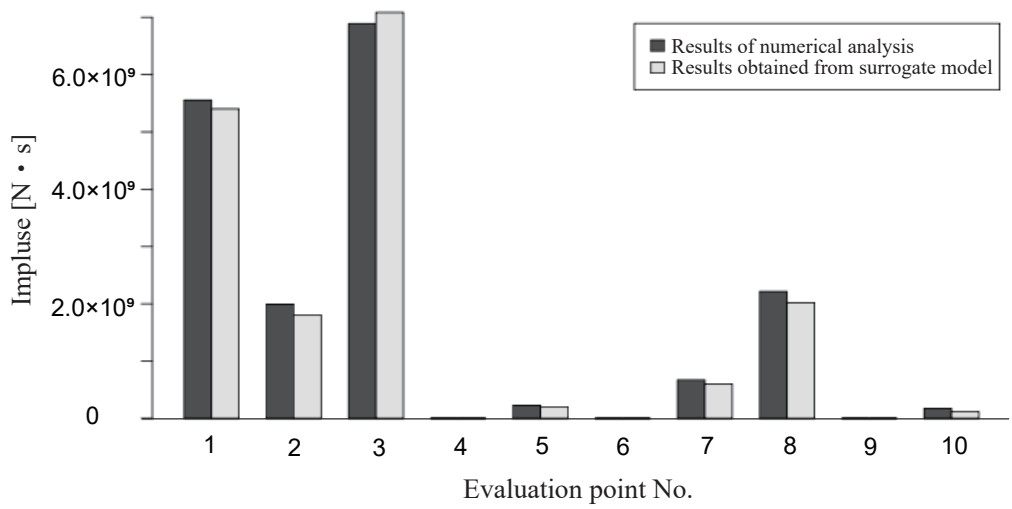

**Figure 29.** Comparison of impulses calculated from the results of numerical analysis and the results obtained from the surrogate model (S5R7)





Here, it is confirmed that it is possible to represent the simulation results using arbitrary parameters with the surrogate model. In concrete terms, the error rate is calculated. For each of the physical quantities, the mean squared error for the time-series data of all points are calculated. The results are shown as follows in Table 2.

**Table 2.** Error between the results of the numerical analysis and the results obtained from the surrogate model

| case name | error [N] (impact force) | error [m] (inundation) |
|-----------|--------------------------|------------------------|
| S2R3      | 66930                    | 0.326                  |
| S5R7      | 84017                    | 0.404                  |

With regard to Table 2, the ratio of the mean values was $434\%$ for the impact force data and $414\%$ for the water depth data in S2R3 and $318\%$ for the impact force data and $252\%$ for the water depth data in S5R7. It is acknowledged that these are very high values. This large error comes from inadequate representation of local peaks and the phase shift. In addition, because the time-series data includes a large amount of data that would not be affected by a tsunami, such as data from mountainous areas, the data include many values equal or close to zero: this would result in extremely small mean values. That is, it is likely that the large error rate value in calculations is because of the small mean values. Note that the ratio of the maximum values is $0.78\%$ for the impact force data and $2.15\%$ for the water depth data for S2R3 and $0.78\%$ for the impact force data and $2.04\%$ for the water depth data in the case of S5R7. In these cases, the error rate is low compared to the maximum value. In other words, because the error in Table 2 is sufficiently small compared to the maximum value, the overall distribution and the general shape of the time-series data can be roughly captured by the surrogate model.

Next, as in the previous section, the impulse is calculated using the time-series data of the impact forces, and a comparison of the impulse calculated from the numerical analysis results and the results obtained from the surrogate model for the 10 evaluation points is shown in Figs. 28 and 29.

Figs. 28 and 29 show that the surrogate model can adequately represent the impulse calculated from the numerical simulation results. Since the impulse can visualize the risk that cannot be seen by the maximum impact force alone, it can be fully utilized for the tsunami risk assessment for buildings in the target area.

Furthermore, the surrogate model has distinct advantages in terms of calculations costs. Four hours was required to complete the numerical analysis of one case when using Intel(R) Xeon(R) CPU E5-2667 v4 (3.20 GHz) 16 parallel computing for the 2D calculation. In the case of the 3D analysis, using an Intel Xeon Phi KNL (1.4 GHz) 8 node 544 core, the calculation time was approximately 96 hours. Once the model was constructed, however, the calculation using the surrogate model took only a few seconds, and because it was possible to calculate the spatial temporal distribution of the physical quantity using arbitrary parameters, it would be possible to apply the surrogate model using the spatial modes for real-time damage prediction.




## 4    Conclusions

This study presents the framework for predicting time series data of tsunami force based on the results of a high precision

numerical analysis at a low calculation cost. By performing proper orthogonal decomposition in relation to the numerical
analysis results while considering uncertainty, we could extract spatial modes and express the spatial distribution of the risk
indicators as a linear distribution of the modes. Additionally, by expressing the coefficients as functions of analysis parameters
and time in the surrogate model, it was possible to calculate the distribution at extremely low calculation costs for certain cases
for which no numerical analysis has been performed. In this study, surrogate models of the impact force and inundation depth

of a tsunami running up to a target area were constructed, and it was shown that the results of the numerical analysis could be
roughly represented. It is also shown that the impulse calculated from the time-series data of the tsunami force obtained from
the surrogate model can sufficiently represent the numerical simulation results. These results indicate that the surrogate model
can be efficiently utilized for tsunami risk assessments.

In this study, for tsunami events, only the two fault parameters of slip and rake was considered as uncertainties. However,

as many uncertainties are at play in an actual tsunami event, it would be possible to integrate these parameters and perform
a numerical analysis and, in the same manner, create a surrogate model that incorporates a larger range of uncertainties in
more detail. However, when increasing the types of input parameters, as the number of scenarios that need to be considered is
immense, it is important to use an efficient method for setting the scenarios (McKay (1979)). Furthermore, it is necessary to
evaluate in advance the extent to which these uncertainty parameters will fluctuate. In the surrogate model created in this study,

a sufficient evaluation was performed with interpolation (within the scope of the parameters for which numerical analysis was
conducted). However, with extrapolation (outside the scope of the parameters for which numerical analysis was conducted), the
possibility of reduced accuracy is high, so it is necessary to evaluate in advance the extent to which the uncertainty parameters
fluctuate and set scenarios that allow for this range of fluctuation.

While the mechanism shown in this paper was designed for use with tsunamis, it has potential for use in the real-time simu-

lation of various disasters. Since numerical simulations of disasters typically involve high computational costs, it is important
to consider uncertainty in advance and perform a numerical analysis before creating a surrogate model. That is, the real-time
simulations developed by the surrogate model proposed in this work can be used as a tool to gauge the extent of disasters.

## Appendix A:  Stabilized finite element method

With regard to Eqs. (4) and (5), we apply the stabilized finite element method equipped with the streamline-upwind/Petrov-

Galerkin (SUPG) and pressure-stabilizing/Petrov-Galerkin(PSPG) (SUPG/PSPG method) (Tezduyar, 1991) to obtain the fol-




lowing finite element equation:

$$
\rho \int_{\Omega_{\text{ns}}} \boldsymbol{w}^h \cdot \rho \left( \frac{\partial \boldsymbol{u}^h}{\partial t} + \boldsymbol{u}^h \cdot \nabla \boldsymbol{u}^h - \boldsymbol{f} \right) d\Omega \quad + \int_{\Omega_{\text{ns}}} \boldsymbol{\varepsilon}(\boldsymbol{w}^h) : \boldsymbol{\sigma}(\boldsymbol{u}^h, p^h) d\Omega + \int_{\Omega_{\text{ns}}} q^h \nabla \cdot \boldsymbol{u}^h d\Omega
$$

$$
+ \sum_{e=1}^{n_{\text{el}}} \int_{\Omega_{\text{ns}}^e} \left\{ \tau_{\text{supg}}^{ns} \boldsymbol{u}^h \cdot \nabla \boldsymbol{w}^h + \tau_{\text{pspg}}^{\text{ns}} \frac{1}{\rho} \nabla q \right\} \quad \cdot \left\{ \rho \left( \frac{\partial \boldsymbol{u}^h}{\partial t} + \boldsymbol{u}^h \cdot \nabla \boldsymbol{u}^h - \boldsymbol{f} \right) - \nabla \cdot \boldsymbol{\sigma}(\boldsymbol{u}^h, p^h) \right\} d\Omega
$$

$$
+ \sum_{e=1}^{n_{\text{el}}} \int_{\Omega_{\text{ns}}^e} \tau_{\text{cont}}^{\text{ns}} \nabla \cdot \boldsymbol{w}^h \rho \nabla \cdot \boldsymbol{u}^h d\Omega = 0 \tag{A1}
$$

Here, $\Omega_{\text{ns}} \in \mathbb{R}^3$ is the analysis domain, $n_{\text{el}}$ is the number of elements, $\boldsymbol{u}^h$ and $p^h$ are the finite element approximations for velocity and pressure, respectively, and $\boldsymbol{w}^h$ and $q^h$ are the approximations of the weight functions for the equations of motion and continuity, respectively. Also, the fourth involve the SUPG term for stabilizing the advection process and the PSPG term for avoiding pressure oscillation. In addition, the fifth term has the shock-capturing term (Aliabadi and Tezduyar, 2000) for avoiding numerical instability on free surfaces, and $\tau_{\text{supg}}^{\text{ns}} \tau_{\text{pspg}}^{\text{ns}}$ and $\tau_{\text{cont}}^{\text{ns}}$ are all stabilization parameters.

In this study, the phase-field approach is employed to capture the complex geometries of free surfaces, such as breaking waves and flow around buildings. In this approach, the locations of free surfaces can be determined by solving the Allen-Cahn advection equation (Chiu and Lin, 2011; Takada et al., 2013), which was modified to the storage format as

$$
\frac{\partial \phi}{\partial t} \mathbf{u} \cdot \nabla \phi = \frac{\epsilon}{\delta} \nabla \cdot (\delta(\nabla \phi) - F_a), F_a = \phi(1 - \phi) \frac{\nabla \phi}{|\nabla \phi|} \tag{A2}
$$

where the phase-field function $\phi$ takes 1.0 or 0.0 to identify liquid (water) and gas (air) phases, and an intermediate value represents the free surface. Here, $\varepsilon$ is the mobility and $\delta$ is the representative width of the free surface. According to the phase field function, the fluid density $\rho$ and viscosity coefficient $\mu$ for each element were evaluated as

$$
\rho = \rho_{\text{l}} \phi + \rho_{\text{g}}(1 - \phi) \tag{A3}
$$

$$
\mu = \mu_{\text{l}} \phi + \mu_{\text{g}}(1 - \phi) \tag{A4}
$$

where subscripts $l$ and $g$ indicate the quantities of the liquid and gas, respectively.

In this study, we also apply the stabilized finite element method with the SUPG method to the Allen-Cahn equation above, so that the resulting finite element equations are given as

$$
\int_{\Omega^P} \phi_*^h \left( \frac{\partial \phi^h}{\partial t} + \boldsymbol{u}^h \cdot \nabla \phi \right) d\Omega + \int_{\Omega^P} \epsilon \nabla \phi_*^h \cdot \nabla \phi d\Omega + \int_{\Omega^P} \phi_*^h \frac{\epsilon}{\delta} \nabla F_a d\Omega
$$

$$
+ \sum_{e=1}^{n_{\text{el}}} \int_{\Omega_e^P} \left( \tau_\phi \, \boldsymbol{u}^h \cdot \nabla \phi_*^h \right) \cdot \left( \frac{\partial \phi^h}{\partial t} + \boldsymbol{u}^h \cdot \nabla \phi^h - \frac{\epsilon}{\delta} \nabla \cdot (\delta(\nabla \phi) - F_a) \right) d\Omega = 0 \tag{A5}
$$





where $\phi^h$ and $\phi^h_*$ are finite element approximations for the phase-field function $\phi$ and its weight function, respectively. Here, $\tau_\phi$ is the stabilization parameter defined as

$$\tau_\phi = \left[ \left( \frac{2}{\Delta t} \right)^2 + \left( \frac{2||\boldsymbol{u}^h||}{h_e} \right)^2 \right]^{-\frac{1}{2}} \tag{A6}$$

where $h_e$ is the characteristic length of an element.

*Acknowledgements.* The authors would like to gratefully acknowledge the support made through Chubu Electric Power (Public Research

Concerning Nuclear Power)



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
