# Peer review of "Rapid Tsunami Force Prediction by Mode Decomposition -Based Surrogate Modeling"

_Natural Hazards and Earth System Sciences, 2021_

## Community Comment (CC2)

> This paper contains a very interesting analysis and you have obtained some impressive results. Here are a few comments / suggestions that I hope might be useful.
>
> There were a few things I was confused about regarding the mathematical description of the algorithm that perhaps you could clarify:

Thank you for your valuable comments. Our responses are summarized below.

> In forming the covariance matrix (9), I guess you are assuming that the columns of X have already had their means subtracted?

As the reviewer pointed out, it is sometimes better to subtract the means from the columns of the data matrix. However, we did not subtract the means this time, because there was no difference in the cases with consideration of subtracting the means and without it. Since the point should have been clarified, we would like to mention it in the revised manuscript.

> In line 121, j=1,...,n should be j=1,...,N I think? and also in the summation in the denominator of the equation on line 126 it should be N not n.

Because the number of the rows of the data matrix $\boldsymbol{X}$ is $n$, the size of the covariance matrix is $n \times n$ as shown in Equation (9). This means $n$ eigenvalues can be obtained. Thus, the expression, $j = 1, ...n$, is correct.

> In line 138 where you say C and C' have common eigenvalues, I think in general they have a different number of eigenvalues, but any excess ones are all zero. You never make it clear how many rows the matrix X has in this section, i.e. the number of observations, but I guess this is greater than N in general so that C is a larger matrix than C'?

As the reviewer pointed out, C and C' have a different number of eigenvalues. We should have stated that they have common non-zero eigenvalues. We would like to revise that way.

Regarding the sizes of C and C', as you mentioned, C is generally larger than C' since the length of data vector n is generally larger than the number of scenarios N. Because the meaning of the row of the matrix X was not clear in the manuscript, we would like to add an explanation in the revised manuscript.

> In line 156 do you mean only the coefficients $\alpha_{ij}$ for j=1,..,r not for j=1,...,p ? I guess so from (16), but worth clarifying. So $f(\beta)$ in general is a vector of length r.

The reviewer's inference is correct. As stated after Equation (15), we extract only significant modes from n eivenvalues/vectors to construct the surrogate model, which can therefore be a sort of reduced order model. As the reviewer suggested, we should have clarified this truncation when we wrote Equation (16).

Let us add some appropriate sentences to explain the procedure.

> In this paragraph why do you use subscript j rather than k as used in (15) and (16)?

As the reviewer points out, it is indeed confusing. To avoid unnecessary confusion, we have decided to

replace $f_j(\boldsymbol{\beta})$ by $f_k(\boldsymbol{\beta})$ in Section 2.3 and accordingly revise some parts in this section. In response to this revision, we would like to revise $\alpha_{ij}$ in section 2.3 to $\alpha_{ik}$ because it is more appropriate.

> In (17), I think the exp(...) after the second = sign should be $= \sum_{i=1}^{N} w_i \exp(...)$

The reviewer is absolutely right. Equation (17) in the original manuscript is incorrect as it lacks some symbols. We would like to fix it.

> At first glance (18) seems to be a square N by N linear system so it seems no regression is needed, so maybe it's worth pointing out that it is really Nr by N since $f(\beta)$ has length r.

Equation (18) is a square N by N linear system. But, as stated below this equation, we used ridge regression (Hoerl and Kennard, 1970) to minimize the error of the surrogate models to prevent the overfitting problem. Nevertheless, the explanation seemed to be insufficient. We would like to add an explanatory equation and some additional explanations in the Subsection 2.3 to avoid confusion.

> Lines 188-190 weren't clear to me. Maybe say the slip was varied from 0.7 to 1.4 times the original slip in the model of Figure 2. When I first read this I also thought you were varying the rake between -20 and +25 degrees, which would be wrong for a subduction event, so maybe also make it even clearer that these are the range of perturbations in the rake angle from the ones given by Fujii-Satake?

We determined the ranges of the fault parameters by following the calculation condition reported by Kotani et al. (2020). According to the paper, JSCE (Japan Society of Civil Engineers (2011, in Japanese) conducted calculations with different rake values and reported that ±10 is a suitable rake range. But Kotani et al. (2020) changed the rake from -20 and +25 degrees to cover a more suitable range and check the effect of the variation of rake. Hence, the range of the rake used in our study may not be based on the real perturbations suggested in the Fujii-Satake model. Nevertheless, because the main objective of our study is to propose the instant prediction of tsunami forces, we think the wider range of the rake is not necessarily an irrelevant condition. We ask for the reviewer's understanding.

> In Figure 5 the cyan lines for the Obs. are very hard to see, maybe make these lines black or red?

As the reviewer suggested, we would like to change colors of these lines to discriminate them.

> Line 209, by "concrete connection method" I think you mean the method for coupling the 2D and 3D methods together, but this was confusing to me at first. Maybe say something like "To couple the 2D and 3D models together, the method used in the study of Takase et al (2016) was employed." (By "concrete" I think you mean the specific method employed here, and it might be better to use "specific" here and some other places. Since in English concrete is also a building material, and you are talking about forces on buildings, there could be some confusion.)

Certainly, "concrete" also means a building material and so may be not appropriate here as an adjective. We would like to replace "concrete" by "specific" as suggested by reviewer.

> Line 231, discussing the 2D mesh used for evaluating the tsunami force: do you average (or sum?) the force over all vertical building faces that happen to lie in the 10m cell? It seems like this would vary a lot from cell to cell just based on the particular geometry of the buildings. In particular some cells might include no walls and hence have 0 force (?) while neighboring cells might have one wall or perhaps at the corner of a building a cell has two walls. So I'm surprised the plots of forces look as smooth as they do and perhaps you can say more about this?

The force is calculated by integrating the pressure acting on all the vertical faces of building within each evaluation cell. Therefore, as the reviewer pointed out, the tsunami force strongly depends on the surface area of buildings, and cells with no walls have zero force. Although we might have been able to employ other risk indices, such as average pressure, the force was considered to be the simplest to check the result. Nevertheless further insight into this aspect is left to future work.

> It is great that you can get a surrogate model that reproduces the time evolution as well as it does, in addition to the spatial patterns. But I wonder if this is mainly because you are only considering perturbations to the Fujii-Satake model in which the basic spatial structure of the fault slip is always the same and so the time evolution shows similar sets of waves and arrival times, just somewhat varying magnitudes? The results are still impressive, but I wonder if you can comment on how this might be extended to developing a surrogate model that could be useful in real time for some earthquake that is not a small perturbation of 2011 (which the next big one almost certainly won't be).

As the reviewer pointed out, one of the reason why we could reproduce the time evolution using spatial modes would be that the basic spatial structure of the fault is always the same. Although the range of the rake is wider than the realistic one, as the reviewer suggested, we might be able to check the potential of the surrogate models by considering wider range of the slip. We determined the ranges of the fault parameters by following Kotani et al. (2020), we would like to consider the condition with larger perturbation. Basically, the surrogate models can work even under large perturbation if enough calculation cases are available. But, even in that case, we have to consider the effective parameter sampling. Because the discussion is very important, we would like to add some explanation in the manuscript. In addition, this remains to be explored in future studies.

> To develop a surrogate model that would handle a greater variety of quakes, perhaps it would be necessary to give up on trying to model the full time-dependent solution and instead build a surrogate model that only attempts to predict the maximum inundation depth and force, which might be much easier to do and still very useful.

Yes, we can construct the surrogate model for the maximum value in the same manner and it is also very useful. In fact, although it is not shown in the manuscript, we have constructed it in preliminary studies. But if we can consider both the space and time as in the manuscript, the resulting surrogate model can contain various information and of course accommodates the distribution of the maximum value.

> I don't understand some of the discussion in the paragraph just below Table 2 (lines 312-320). You

say "the ratio of mean values was 434%...". I think a ratio should just be a number, not a percent. Do you mean the ratio was 4.34? And what mean values is this the ratio of? Is it the ratio of the error to the true result, i.e. the relative error? This isn't clear.

Maybe this would be clearer if you included also tables of the raw numbers you are comparing, it's not clear where these numbers come from or how they relate to Table 2.

In line 316-317 you give ratios like 0.78%. Again I'm not sure what you mean by a ratio as a percent, do you mean the ratio of error to true value is 0.0078?

As the reviewer pointed out, the explanation of the error indicated in Subsection 3.2.4 was not appropriate. It might be difficult to understand. We would like to modify the corresponding expressions and Table 2 so that potential reader can easily understand the results.

In spite of my questions and possible confusion, in general I like the paper and believe it should be published after some clarifications.

We appreciate the referee for taking time to review our manuscript with some flaws. Thanks to the reviewer's comments and suggestions, the quality of our paper would be improved significantly after adequate revision.

---

## Author Comment (AC2)

Thank you for your valuable comments. Our responses are summarized below.

> This paper examines how tsunami impact can be predicted rapidly using mode decomposition of the results from 2D (shallow-water) propagation modelling coupled with 3D (Navier-Stokes) inundation modelling. The surrogate model with mode decomposition reproduces quite well the time-series and maps of run-up and impulses. The mini work-flow of how the mode decomposition surrogate model is clearly presented in Figure 1. What is a little more unclear to me is the overall workflow and purpose of the work. The title uses the term "Real-time" which, in my world, is reserved for urgent or situation computations where the process is initiated by a real-world trigger (observational data or human intervention) and that the computations are running against a real-world clock. Is this the case? Otherwise, "Rapid" is probably a better term – that indicates fast computation.

Our final goal is to predict the damage caused by tsunamis in the city using real-time observational data and to provide the extent of damage as soon as possible. However, as pointed out by the reviewer, such a prediction is not synchronized with the tsunami events and rather a rapid action in the limited condition. So, we would like to consider the use of the term "Rapid" instead of "real-time", including the title.

> With regards to the total workflow, is it the intention that the full numerical simulations are calculated beforehand and the surrogate model saved for application when a new tsunami event occurs? Or are all the calculations performed in a new event and the surrogate model used to interpolate the outcome to different parts of the parameter space to those calculated in the 2D/3D simulations?

In the framework proposed in this study, an expected limited number of scenarios are obtained beforehand by conducting a series of numerical simulations with selected sets of fault parameters, and then a surrogate model is constructed based on those results. Then, when an actual tsunami event occurs, the actual fault parameters are given so that the POD coefficients in the surrogate model can be interpolated in the parameter space.

> In any case, there will be a significant variability of the tsunami impact as the source parameters vary (c.f. Table 1) – how is the decision made as to which are the appropriate parameters to be looking at when interpolating the predicted impact using the surrogate model? Is it by real-time comparison between observations and predictions?
> Answering the above questions in the paper would help enormously in making the context of these calculations clear.

As explained above, this study stands on the assumption that the fault parameters are given when an actual tsunami event occurs. Because this point is not clearly mentioned in the manuscript, we would like to add the explanation in the revised manuscript.

> What exactly do the "data vectors" in Equation (8) contain? ("data arranged according to a certain rule") – is it wave heights at the different grid points? Velocities? It would be useful to know which values are stored for each grid point (h,ux,uy?)

The data vector in Equation (8) contains the physical quantities that are used to construct a surrogate model. Specifically, in this study, the tsunami force and inundation depth at each grid point are contained in the data vector. In order to describe the point clearly, we would like to add an more clear and specific explanation around Equation (20).

> Can you comment on the boundary between the 2D analysis and 3D analysis?
>
> It is typical to specify a given water depth at which the 3D-analysis would take over but the line indicated in Figure (6) cuts across a bay very close to the inundation area with very shallow water on each side. How does the transition from 2D to 3D happen on such a boundary? Would it have been feasible to take the boundary further out to sea?

The time history data of two-dimensional simulation results, that are specifically the water depth and average flow velocity, on the boundary are stored and transferred to the three-dimensional numerical analysis. Since the time interval in the 2D calculation differs from that in the 3D one, the 2D results are linearly interpolated in the time domain, and the interpolated values are given to the 3D analysis as input data.

Regarding the interface between the 2D and 3D domains, as pointed out by the reviewer, there might be a suitable position in more offshore area. However, this would increase the 3D analysis area and at the same time increase the computational cost. Therefore, we selected the current position of the boundary line for feasible 3D analysis. As need arises, some comments on this might be added somewhere in the revised manuscript.

> In Figure 8, it would be helpful to have an indication of scale on each of the panels. Is each panel a zoom-in of the previous panel? Does Figure 12 show us something fundamentally different to Figure 8? If so, it would be very valuable to know what is fundamentally different. It looks like there are triangular elements in Figure 12 but not in Figure 8. Is this significant?

Thank you for pointing out the flaw in Figure 8. We would like to add the scale in Figure 8 as suggested.

Regarding the difference between Figs 8 and 12, we would like to say as follows: Figure 8 shows the overall image of the mesh of the target area, while Figure 12 shows 10-meter mesh that is generated to define the sub-area in which the tsunami force is evaluated. Both of them are the same, but the latter is shown to illustrate how the tsunami force is evaluated. Please note that the evaluation of tsunami force is dome by averaging in each of these sub-domains but not for each building. Hence, this is significant. Since the explanation for the structured grid in Figure 12 seems to have been insufficient, we would like to improve the description about it in the revised manuscript.

> In Figure 7, the colour scheme is a little unfortunate with low-lying areas coloured in blue. Whenever I see this Figure, I assume that I am seeing tsunami inundation with the blue areas representing the region with inundation. Would it be possible to have blue at sea level and below and non-blue for the region above land – or at least a clear line indicating the pre-tsunami coastline?
>
> The confusion continues into Figure 9 where I guess it is the white which represents the inundation.

As pointed out by the reviewer, the blue area may look like a tsunami inundation area. We would like to modify the color in Figs 7 and 9 so that they would not give an erroneous impression.

> It would be nice to have the link to the inundation height observations in the caption to Figure 10. This is for the journal to decide I guess.

We would like to add the information about the website in the caption.

> Is the data matrix X in Equation (19) the same as the data matrix X in Equation (8)? I am guessing not as I see the matrix in Equation 8 being a spatial discretization of simulation parameters (time-dependent or not time-dependent?) Is the matrix X time-series for a single metric at one point evaluated for different slip and rake as a function of time? What about the X in Equation (8)? This is something evaluated for many points. I think all of this needs clearing up.

The matrix in equation (19) is different from that in equation (8). The equation (19) is a matrix containing the time series data for one scenario and then the data matrix is defined as shown in equation (20) that contains all the scenario data. Therefore, Equation (8) corresponds to Equation (20).

Note that, in the case of time-space mode decomposition, a data matrix is defined by storing the time series data obtained for all the scenarios in the column direction and separately prepared according to the parameter variation. Because this point was not adequately mentioned in the original manuscript, we would like to add the explanation around the equations (8), (9), and (20).

> What is the quantity we are seeing in Figure 13? It goes from 1 to -1 – it is a fully-normalized data vector? So there is no direct physical interpretation of these numbers? This should be made clear in the figure caption.

Figure 13 shows a normalized representation of the spatial modes. As for the mode numbers, the low-order modes have high contributions with respect to the original data (simulated results) and high-order modes express minor effects. Because the physical meanings are not sufficiently explained in the original manuscript, we would like to add the explanation.

> There are very many figures and I think a lot of care needs to be taken to make it clear in the caption what is different for each figure from similar figures. (e.g. Figure 17 has a map with the locations of evaluation points and we do not see until Figure 26 where these are applied. There are 29 figures in total and I would ask as to whether all are necessary. The reader struggles to understand the significance of each of them. (e.g. Figures 28 and 29 are almost identical – we get the point.)

We would like to reconsider the arrangements of the figures and their captions to make them more easily understand for potential readers.

---

## Author Comment (AC3)

Thank you for your valuable comments and suggestions. Below, we first cite them for reference and make our responses.:

Comments on "Real-time Tsunami Force Prediction by Mode Decomposition-Based Surrogate Modeling" by Kenta Tozato et al. submit to the journal of NHESS. The authors tried to estimate the hydrodynamic force of tsunami acting on building through 2D and 3D FEM. The paper is interesting, and from my personal point of view, the topic of this manuscript fits well with the scope of the journal of NHESS. There are too many figures (snapshots), however few explanations and not ready. Animations or supplementary files may be attached to improve readability. Some other major issues are listed below:

We appreciate your suggestion to provide animations or supplementary files. That is a good idea. Let us consider it.

1. One faulting model of Tohoku Earthquake 2011 is used in this study, however but it may not (and should not) be reactivate at the same location and did not have the same magnitude. So it cannot be applied to other events, nor can it "predict" damages. Therefore, "real-time" tsunami force predictions may not be useful because the events of 2011 have already occurred. The term "real time" may change with the reassessment of the power of the tsunami...

2. The method proposed by the authors is a time-consuming task and not taking into account the faulting parameter of individual event. Likewise, the term "real-time" may not be a suitable term.

Since these two comments suggest that the term "real-time" be changed to a more appropriate one, let us answers to them at once.

As you pointed out, it takes a lot of time to perform numerical simulations for various scenarios. However, all the simulations are conducted in advance to construct a surrogate model. Once the model is completed, the real-time prediction is ready. That is, when an actual tsunami occurs and its fault parameters are determined in some way, the present surrogate model can predict not only inundation heights and areas, but also hydrodynamic forces of tsunami acting on buildings within a very short period of time. Although we applied the proposed method to the Tohoku Eearthquake 2011 in this study, the framework has broad utility and therefore is capable of estimating risks of other events.

Nevertheless, the term "real-time" may not be suitable for this paper. Indeed, similar comments have been left by other reviewers. In response, we would like to consider the use of the term "Rapid" or "Instant" instead of "real-time" in the revised manuscript. We ask for your understanding.

3. There are too many figures, but similar···, need to be simplified. The wording of the manuscript should be further elaborated so that the reader can understand what the author wants to express, regarding to the figures.

4. Many figures is not ready, and some parts can be unified. E.g. Fig 6, 7, 8, 11, 17 etc.

These two comments also suggest the same amendment and therefore we answer to them at once.

As you pointed out, there may be unnecessarily many pictures in a single figure and some of them can be reduced. For example, although there four pictures in Figure 18(a), two might be enough to illustrate

what we want to explain. Similarly, we would like to consider the reduction of the number of pictures in each of Figure 19, 20, 22, 23, 24, 25, 26, 27.

In addition, since some captions may be too simple to make our intentions clear, we would like to add some explanations so that potential readers easily can understand them.

Moreover, as you pointed out, some of the figures can be combined or merged. For example, Figures 6, 11 and 17 can be combined. Also, Figures 7 and 9 can be merged so that the number of pictures can be reduced.

All of these modifications will be made in the revised manuscript.

---

## Author Response (AR1)

**Responses to review comments**

We deeply appreciate reviewers' valuable comments and kind suggestions. We have revised the manuscript in line with the review comments. Responses to reviewers' comments are summarized below. The revised parts are colored in red in the following responses and the revised manuscript.

**#RC1 2021/5/17**

**RC 1-1**

> How do you consider the impact of something being destroyed, such as a levee breach? If that impact is not considered, I don't think any good surrogate model can make predictions!
> https://nhess.copernicus.org/preprints/nhess-2021-77/#RC1

This study mainly aims to present a framework for instant tsunami prediction using a reliable surrogate model and to demonstrate its potential through numerical examples. As you pointed out, it is important for the realistic risk evaluation of tsunami run-up to consider the impact of something being destroyed. However, as you may know, a levee breach and the failure of buildings caused by the tsunami force are still challenging research topics, since relevant numerical methods have not been fully developed yet. In addition, even if we had them at hand, the results would be unreliable in most actual situations because the input parameters and analysis conditions involve lots of uncertainties. If the validity of the calculation results cannot be guaranteed, the corresponding surrogate model does not make sense. Therefore, we think that this is hardly the time for considering the effects of the failure in the proposed framework. However, because we also believe that the effect is important for realistic risk evaluation, The point are hence left for future work.

RC 2-1

> This paper examines how tsunami impact can be predicted rapidly using mode decomposition of the results from 2D (shallow-water) propagation modelling coupled with 3D (Navier-Stokes) inundation modelling. The surrogate model with mode decomposition reproduces quite well the time-series and maps of run-up and impulses. The mini work-flow of how the mode decomposition surrogate model is clearly presented in Figure 1. What is a little more unclear to me is the overall workflow and purpose of the work. The title uses the term "Real-time" which, in my world, is reserved for urgent or situation computations where the process is initiated by a real-world trigger (observational data or human intervention) and that the computations are running against a real-world clock. Is this the case? Otherwise, "Rapid" is probably a better term – that indicates fast computation.

Our final goal is to predict the damage caused by tsunamis in the city using real-time observational data and to provide the extent of damage as soon as possible. However, as pointed out by the reviewer, such a prediction is not synchronized with the tsunami events and rather a rapid action in the limited condition. To express this point clearly, we have replaced the term "real-time" by "Rapid" throughout the manuscript, including the title.

[Original manuscript, page 1, title]
Real-time Tsunami Force Prediction by Mode Decomposition -Based Surrogate Modeling
[Revised manuscript, page 1, title]
Rapid Tsunami Force Prediction by Mode Decomposition -Based Surrogate Modeling

[revised manuscript text omitted]

> With regards to the total workflow, is it the intention that the full numerical simulations are calculated beforehand and the surrogate model saved for application when a new tsunami event occurs? Or are all the calculations performed in a new event and the surrogate model used to interpolate the outcome to different parts of the parameter space to those calculated in the 2D/3D simulations?

In the framework proposed in this study, we carry out a series of numerical simulations with selected sets of fault parameters corresponding to expected scenarios beforehand. Thus, the number of selections are limited and determined according to the degrees of accuracy and efficiency for constructing a surrogate model. Once all the simulation results are obtained, a surrogate model is constructed following the procedure described in Section 2. When an actual tsunami event occurs, the actual fault parameters are given, so that the POD coefficients in the surrogate model can be interpolated in the parameter space. Since the response is related to the next comment, it is explained in the next response.
* * *
RC 2-3

> In any case, there will be a significant variability of the tsunami impact as the source parameters vary (c.f. Table 1) – how is the decision made as to which are the appropriate parameters to be looking at when interpolating the predicted impact using the surrogate model? Is it by real-time comparison between observations and predictions?
>
> Answering the above questions in the paper would help enormously in making the context of these calculations clear.

As explained above, this study stands on the assumption that the fault parameters are given when an actual tsunami event occurs. Because this point was not clearly mentioned in the original manuscript, we have added the explanation in the revised manuscript as follows:

[Original manuscript, page 3, line 72-76]

This section describes a flow and methodologies of the proposed framework. As mentioned in previous section, principal spatial modes are extracted from precomputed simulation data that are obtained from 2D-3D coupled tsunami analysis, and the surrogate models are constructed using the spatial modes for the real-time tsunami force prediction. Fig.1 shows the flowchart of the proposed method. Each parts of the proposed method, such as tsunami analyses, mode decomposition, and surrogate modeling, are explained in details in the following subsections.

[Revised manuscript, page 3, line 73-81]

This section describes a flow and methodologies of the proposed framework. In the framework proposed in this study, we carry out a series of 2D-3D coupled tsunami analyses with selected sets of fault parameters corresponding to expected scenarios beforehand to obtain the limited number of scenario-specific simulation results. Then, the principal spatial modes of tsunami forces are extracted from the precomputed simulation data to construct the surrogate models for the rapid tsunami force prediction. It is to be noted that the coefficients of the modes are interpolated in the parameter space so as to be functions of fault parameters. Thus, because this study stands on the assumption that the fault parameters are given when an actual tsunami event occurs, the tsunami force prediction is made immediately using the constructed surrogate model. Fig.1 shows the flowchart of the proposed method. Each parts of the proposed method, such as tsunami analyses, mode decomposition, and surrogate modeling, are explained in details in the following subsections.

**RC 2-4**

> What exactly do the "data vectors" in Equation (8) contain? ("data arranged according to a certain rule") – is it wave heights at the different grid points? Velocities? It would be useful to know which values are stored for each grid point (h,ux,uy?)

The data vector in Equation (8) contains the physical quantities that are used to construct a surrogate model. Specifically, in this study, the tsunami force and inundation depth at each grid point are contained in the data vector. In order to describe the point clearly, we have added an more clear and specific explanation around Equation (20).

[Original manuscript, page 16, line 250]

Here, $\boldsymbol{x}(U, \lambda, t)$ is a vector showing the spatial distribution of the risk indicator in relation to time.

[Revised manuscript, page 16, line 265-267]

Here, $\boldsymbol{x}(U, \lambda, t)$ is a vector containing the spatial distribution of a risk indicator in relation to time. Specifically, in this study, the tsunami force and inundation depth at each grid point are selected for the indicators and their time-series data are stored in separate data matrices, each of which consists of the data vectors at each time.

**RC 2-5**

> Can you comment on the boundary between the 2D analysis and 3D analysis?
>
> It is typical to specify a given water depth at which the 3D-analysis would take over but the line indicated in Figure (6) cuts across a bay very close to the inundation area with very shallow water on each side. How does the transition from 2D to 3D happen on such a boundary? Would it have been feasible to take the boundary further out to sea?

The time history data of two-dimensional simulation results, which are specifically the water depth and

flow velocity, on the boundary are stored and transferred to the three-dimensional numerical analysis. Since the time interval in the 2D calculation differs from that in the 3D one, the 2D results are linearly interpolated in the time domain, and the interpolated values are given to the 3D analysis as input data.

Regarding the interface between the 2D and 3D domains, as the reviewer pointed out, there might be more suitable position in offshore area. However, even in such a situation, it is necessary to consider the connection between the boundary line and land area. In addition, we have to consider a curved boundary that makes calculation cost increase because 3D analysis domain becomes larger. The advantage of the current boundary position shown in Fig.6 is that the boundary can be a straight line. Therefore, we selected the current position of the boundary line at the mouth of the bay. Because there was a lack of explanation, we revised the manuscript as follows:

[Original manuscript, page 12, line 210-211]
An image of the location of 2D-3D boundary is shown in Fig. 6. The time-series data of the wave height and flow velocity obtained from the 2D wide-area analysis were used as 3D input values.

[Revised manuscript, page 11, line 220-225]
An image of the location of 2D-3D boundary is shown in Fig. 6. The time-series data of the wave height and flow velocity obtained from the 2D wide-area analysis are stored and transferred to the three-dimensional numerical analysis by linear interpolation in space. Also, since the time interval in the 2D calculation differs from that in the 3D one, the 2D results are linearly interpolated in the time domain, and the interpolated values are given to the 3D analysis as input data. The reason why the 2D-3D boundary is placed at the mouth of the bay is that the boundary can be defined by a straight line.

RC 2-6

> In Figure 8, it would be helpful to have an indication of scale on each of the panels. Is each panel a zoom-in of the previous panel? Does Figure 12 show us something fundamentally different to Figure 8? If so, it would be very valuable to know what is fundamentally different. It looks like there are triangular elements in Figure 12 but not in Figure 8. Is this significant?

Thank you for pointing out the flaw in Figure 8. We have added the scale in Figure 8 as suggested.
[Original manuscript, page 14, Figure 8]
Figure 8. Images of FE meshes at different scales.
[Revised manuscript, page 14, Figure 7]
Figure 7. Bird view of FE meshes at two different rates of magnification.

Regarding the difference between Figs 8 and 12, Figure 8 shows the overall image of the mesh of the target area, while Figure 12 shows 10-meter mesh that is generated to define the sub-domains in which the tsunami force is evaluated. We think both figures are needed to illustrate how the tsunami force is evaluated. Since the explanation for the structured grid in Figure 12 was insufficient in the original manuscript, we have improved the description as follows:

[Original manuscript, page 12, line 231-232]

To avoid this problem, we consider a 2D mesh with a grid size of 10m for evaluating the tsunami force. An image of the mesh is shown in Fig. 12.

[Revised manuscript, page 13, line 245-247]

To avoid this problem, we consider a 2D mesh with a grid size of 10m for evaluating the tsunami force. The tsunami force is evaluated by averaging in each of these sub-domains but not for each building in this study. An image of the mesh is shown in Fig. 10.

RC 2-7

> In Figure 7, the colour scheme is a little unfortunate with low-lying areas coloured in blue. Whenever I see this Figure, I assume that I am seeing tsunami inundation with the blue areas representing the region with inundation. Would it be possible to have blue at sea level and below and non-blue for the region above land – or at least a clear line indicating the pre-tsunami coastline?
>
> The confusion continues into Figure 9 where I guess it is the white which represents the inundation.

As pointed out by the reviewer, the blue area may look like a tsunami inundation area. We have deleted Fig. 7, merged the elevation and area size information in Fig. 7 into Fig. 9, and added coastline information to Fig. 9.

[Original manuscript, Figure 7 (page 13) and Figure 9 (page 14)]

Figure 7. 3D analysis area.

Figure 9. Snapshots of tsunami runup obtained in 3D analysis.

[Revised manuscript, Figure 8 (page 14)]

Figure 8. Snapshots of tsunami runup obtained in 3D analysis. The white-colored area represents the inundation area.

RC 2-8

> It would be nice to have the link to the inundation height observations in the caption to Figure 10. This is for the journal to decide I guess.

We have added the information about the website in the caption.

[Original manuscript, page 15, Figure 10]

Figure 10. Comparison of inundation height between observation and numerical simulation.

[Revised manuscript, page 15, Figure 9]

Figure 9. Comparison of inundation heights between observational data and simulation results. (Observation data are provided by the 2011 Tohoku Earthquake Tsunami Joint Survey Group (2012))

RC 2-9

> Is the data matrix X in Equation (19) the same as the data matrix X in Equation (8)? I am guessing not as I see the matrix in Equation 8 being a spatial discretization of simulation parameters (time-dependent or not time-dependent?) Is the matrix X time-series for a single metric at one point evaluated for different slip and rake as a function of time? What about the X in Equation (8)? This is something evaluated for many points. I think all of this needs clearing up.

The matrix in equation (19) is different from that in equation (8). Equation (19) is a matrix containing the time series data for one scenario and then the data matrix is defined as shown in equation (20) that contains all the scenario data. Therefore, Equation (8) corresponds to Equation (20).

Note that, in the case of time-space mode decomposition, a data matrix is defined by storing the time series data obtained for all the scenarios in the column direction and separately prepared according to the parameter variation. Because this point was not adequately mentioned in the original manuscript, we added the explanation around the equations (8), (9), and (20).

[Original manuscript, page 6, line 116-117]
Here, a vertical line is added to each column vector to implicate that the component alignment follows the specified rule. The covariance matrix of the data matrix is defined as ⋯
[Revised manuscript, page 6, line 122-126]
Here, a vertical line is added to each column vector to implicate that the component alignment follows the specified rule. In POD, the data matrix is generally constructed with the mean subtraction, but in this study, the procedure was not applied because the accuracy of the constructed surrogate model with subtraction was almost the same as the one without subtraction. In the case of time-space mode decomposition, a data matrix is defined by storing the data for each time in the column direction. The covariance matrix of the data matrix is defined as ⋯

[Original manuscript, page 17, line 255]
The previously described proper orthogonal decomposition was carried out for this data matrix.
[Revised manuscript, page 16, line 272-274]
In this study, because time-space mode decomposition is performed, the data for each calculation case and each time are aligned in the column direction of the data matrix. The previously described proper orthogonal decomposition was carried out for this data matrix.

RC 2-10

> What is the quantity we are seeing in Figure 13? It goes from 1 to -1 – it is a fully-normalized data vector? So there is no direct physical interpretation of these numbers? This should be made clear in the figure caption.

Figure 13 shows a normalized representation of the spatial modes. As for the mode numbers, the low-order modes have high contributions with respect to the original data (simulated results) and high-order modes express minor effects. Because the physical meanings were not sufficiently explained in the

original manuscript, we have added the explanation as follows:

[Original manuscript, page 17, line 263-267]
From these figures, it can be confirmed that the impact force distribution and inundation depth distribution both have roughly the same spatial distribution characteristics. If we examine the characteristics in more detail, in the case of the first mode, we can see the tendency for the coastal side to be most strongly impacted. Naturally, since the points close to the coast are most affected by the tsunami, this is a mode that best expresses this kind of impact. Additionally, the second mode expresses opposite tendencies for the section close to the coast and the section behind it, and the third mode expresses opposite tendencies for the eastern and western coastal areas.

[Revised manuscript, page 17-18, line 280-287]
The values in Fig. 11 show the normalized ones for each mode. From these figures, it can be confirmed that the impact force distribution and inundation depth distribution both have roughly the same spatial distribution characteristics. If we examine the characteristics in more detail, in the case of the first mode, we can see the tendency for the coastal side to be most strongly impacted. Naturally, since the points close to the coast are most affected by the tsunami, this is a mode that best expresses this kind of impact. The second mode expresses opposite tendencies for the section close to the coast and the section behind it, and the third mode expresses opposite tendencies for the eastern and western coastal areas. In general, the lower-order modes have higher contributions with respect to the original data, while the higher-order modes represent local effects and their contributions are relatively small.

RC 2-11

> There are very many figures and I think a lot of care needs to be taken to make it clear in the caption what is different for each figure from similar figures. (e.g. Figure 17 has a map with the locations of evaluation points and we do not see until Figure 26 where these are applied. There are 29 figures in total and I would ask as to whether all are necessary. The reader struggles to understand the significance of each of them. (e.g. Figures 28 and 29 are almost identical – we get the point.)

Thanks to these comments, we have reconsidered the arrangements of the figures and their captions to make them easier to understand. The following table shows the all of the revisions of figures. The revised captions are also shown in the list. The revisions of figures shown in the responses to RC 2-6, 2-7 and 2-8 are also included in the list. Because some figures have been merged, the figure numbers have been changed after Figure 7.

| Figure numbers in original manuscript | Figure numbers in revised manuscript | Captions |
|---|---|---|
| 1 | 1 | Flowchart of the rapid tsunami force prediction by mode decomposition-based surrogate model. |
| 5 | 5 | Comparison of tsunami heights between observational data and simulation results. (borrowed from Kotani et al. (2020)) |
| 6, 11 | 6 | Boundary between the 2D analysis and 3D analysis areas. Points A to H are used to compare the inundation depths between observational data and simulation results. (© Google Maps) |
| 8 | 7 | Bird view of FE meshes at two different rates of magnification. |
| 7, 9 | 8 | Snapshots of tsunami runup obtained by 3D analysis. The white-colored area represents the inundation area. |
| 10 | 9 | Comparison of inundation heights between observational data and simulation results. (Observation data are provided by the 2011 Tohoku Earthquake Tsunami Joint Survey Group (2012).) |
| 13, 14 | 11 | Spatial modes of (a) impact force and (b) inundation depth extracted by POD. (© Google Maps) |
| 16 | 13 | The root mean squared error for the results obtained from the numerical analysis and those of the surrogate model for each mode. |
| 17 | 14 | Evaluation points for comparing the results obtained from numerical simulations and those of the surrogate model. (© Google Maps) |
| 18, 19 | 15 | Snapshots of the results obtained from numerical analysis and those reconstructed by using spatial modes. (© Google Maps) |
| 20 | 16 | Comparison of time-series data obtained from numerical analysis and those reconstructed by using spatial modes. |
| 22, 23 | 18 | Snapshots of the results obtained from numerical analysis and those obtained from the surrogate model. (S2R3) (© Google Maps) |
| 24, 25 | 19 | Snapshots of the results obtained from numerical analysis and those obtained from the surrogate model. (S5R7) (© Google Maps) |
| 26, 27 | 20 | Comparison of time-series data obtained from the numerical analysis and surrogate model. |
| 28, 29 | 21 | Comparison of impulses calculated from the results of numerical analysis and those obtained from the surrogate model. ((a)S2R3, (b)S5R7) |

**#CC1 2021/8/5**

**CC 1-1**

> This paper contains a very interesting analysis and you have obtained some impressive results. Here are a few comments / suggestions that I hope might be useful.
>
> There were a few things I was confused about regarding the mathematical description of the algorithm that perhaps you could clarify:

Thank you for your valuable comments. Our responses are summarized below.

**CC 1-2**

> In forming the covariance matrix (9) I guess you are assuming that the columns of X have already had their means subtracted?

As the reviewer pointed out, it is sometimes better to subtract the means from the columns of the data matrix. However, we did not subtract the means this time, because there was no difference in the cases with consideration of subtracting the means and without it. Since the point should have been clarified, we have added it in the revised manuscript.

[Original manuscript, page 6, line 116]
Here, a vertical line is added to each column vector to implicate that the component alignment follows the specified rule.

[Revised manuscript, page 6, line 122-124]
Here, a vertical line is added to each column vector to implicate that the component alignment follows the specified rule. In POD, the data matrix is generally constructed with the mean subtraction, but in this study, the procedure was not applied because the accuracy of the constructed surrogate model with subtraction was almost the same as the one without subtraction.

**CC 1-3**

> In line 121, j=1,...,n should be j=1,...,N I think? and also in the summation in the denominator of the equation on line 126 it should be N not n.

Because the number of the rows of the data matrix $\boldsymbol{X}$ is $n$, the size of the covariance matrix is $n \times n$ as shown in Equation (9). This means $n$ eigenvalues can be obtained. Thus, the expression, $j = 1, ... n$, is correct.

**CC 1-4**

> In line 138 where you say C and C' have common eigenvalues, I think in general they have a different number of eigenvalues, but any excess ones are all zero. You never make it clear how many rows the matrix X has in this section, i.e. the number of observations, but I guess this is greater than N in general so that C is a larger matrix than C'?

As the reviewer pointed out, C and C' have a different number of eigenvalues. We should have stated that they have common non-zero eigenvalues. Regarding the sizes of C and C', as you mentioned, C is

generally larger than C' since the length of data vector n is generally larger than the number of scenarios N. Because the meaning of the row of the matrix X was not clearly explained in the manuscript, we have added an explanation in the revised manuscript. Revised parts are summarised below.

[Original manuscript, page 7, line 138-139]
It follows that the $C$ and $C'$ have common eigenvalues, each of which is equal to the squared singular value of $X$, and that $U$ and $V$ in Eqn. (13) correspond to the eigenvectors of $C$ and $C'$ , respectively.

[Revised manuscript, page 7, line 147-148]
It follows that the $C$ and $C'$ have common non-zero eigenvalues, each of which is equal to the squared singular value of $X$, and that $U$ and $V$ in Eqn. (13) correspond to the eigenvectors of $C$ and $C'$ , respectively.

[Original manuscript, page 6, line 112-113]
When $n$ data are obtained for scenario case $i$, they are stored into a column vector denoted by $x_i$, which has n components.
[Revised manuscript, page 6, line 117-119]
When $n$ data are obtained for scenario case $i$, they are stored into a column vector denoted by $x_i$, which has $n$ components. Here, $n$ is the number of evaluation points for risk indicators, namely the tsunami force and inundation depth in this study.

CC 1-5

> In line 156 do you mean only the coefficients $\alpha_{ij}$ for j=1,..,r not for j=1,...,p ? I guess so from (16), but worth clarifying. So $f(\beta)$ in general is a vector of length r.

The reviewer's inference is correct. As stated after Equation (15), we extract only significant modes from n eivenvalues/vectors to construct the surrogate model, which can therefore be a sort of reduced order model. As the reviewer suggested, we have clarified this point.

[Original manuscript, page 7, line 156-157]
That is, coefficients $\alpha_{ij}$ are approximated as functions of parameters $\beta$ by interpolation or regression, which are denoted by $f_j(\beta)$.
[Revised manuscript, page 7, line 165-166]
That is, coefficients $\alpha_{ik}$ $(k = 1, ..., r)$ are approximated as functions of parameters $\beta$ by interpolation or regression, which are denoted by $f_k(\beta)$ $(k = 1, ..., r)$.

CC 1-6

> In this paragraph why do you use subscript j rather than k as used in (15) and (16)?

As the reviewer points out, it is indeed confusing. To avoid unnecessary confusion, we have replaced $f_j(\beta)$ by $f_k(\beta)$ in Section 2.3 and accordingly revised some parts in the section of the revised manuscript.

In response to this revision, we have revised $\alpha_{ij}$ to $\alpha_{ik}$ in Section 2.3 of the revised manuscript because it is more appropriate.

[Original manuscript, page 7, line 156-162]
That is, coefficients $\alpha_{ij}$ are approximated as functions of parameters $\boldsymbol{\beta}$ by interpolation or regression, which are denoted by $f_j(\boldsymbol{\beta})$. Then, the surrogate model can be expressed in the following equation whose independent variables are input parameters $f_j(\boldsymbol{\beta})$:
( Equation (16) )
In this study, $f_j(\boldsymbol{\beta})$ is determined by the interpolation with radial basis functions (RBF) (Buhmann, 1990) because it is applicable even if the parameters are not distributed in equidistant intervals and even if some defects are present in the data.

   RBF interpolation for $f_j(\boldsymbol{\beta})$ can be expressed by the following equation:
[Revised manuscript, page 7-8, line 165-171]
That is, coefficients $\alpha_{ik}$ $(k = 1, ..., r)$ are approximated as functions of parameters $\boldsymbol{\beta}$ by interpolation or regression, which are denoted by $f_k(\boldsymbol{\beta})$ $(k = 1, ..., r)$. Then, the surrogate model can be expressed in the following equation whose independent variables are input parameters $f_k(\boldsymbol{\beta})$:
( Equation (16) )
In this study, $f_k(\boldsymbol{\beta})$ is determined by the interpolation with radial basis functions (RBF) (Buhmann, 1990) because it is applicable even if the parameters are not distributed in equidistant intervals and even if some defects are present in the data.

   RBF interpolation for $f_k(\boldsymbol{\beta})$ can be expressed by the following equation:

**CC 1-7**

> In (17), I think the exp(...) after the second = sign should be = $\sum_{i=1}^{N} w_i \exp(...)$

   The reviewer is absolutely right. Equation (17) in the original manuscript is incorrect as it lacks some symbols. We have fixed it.

[Original manuscript, page 7, line 163 (Equation 17)]
$f(\boldsymbol{\beta}) = \sum_{i=1}^{N} w_i \phi(\boldsymbol{\beta}, \boldsymbol{\beta}_i) = \exp(-\gamma ||\boldsymbol{\beta} - \boldsymbol{\beta}_i||^2)$
[Revised manuscript, page 8, line 172 (Equation 17)]
$f_k(\boldsymbol{\beta}) = \sum_{i=1}^{N} w_i \phi(\boldsymbol{\beta}, \boldsymbol{\beta}_i) = \sum_{i=1}^{N} w_i \exp(-\gamma ||\boldsymbol{\beta} - \boldsymbol{\beta}_i||^2)$    $(k = 1, ..., r)$

**CC 1-8**

> At first glance (18) seems to be a square N by N linear system so it seems no regression is needed, so maybe it's worth pointing out that it is really Nr by N since $f(\beta)$ has length r.

   $f(\beta)$ has length $N$ and Equation (18) is a square $N$ by $N$ linear system. Because $N$ indicates number of calculation cases and the set of weights are individually calculated in each mode. In terms of regression, as stated in the manuscript, we used ridge regression (Hoerl and Kennard, 1970) to minimize the error of the surrogate models and to prevent the overfitting problem while Equation (18) is a square N by

N linear system. Because the point was not explained clearly, we have revised subsection 2.3 to avoid confusion. In the revised manuscript, it's emphasized that the weights $w_i$ and the RBF parameter $\gamma$ are calculated in each mode. Also, we have replaced $f(\beta)$ by $\alpha$ because the description is more adequate.

[Original manuscript, page 7-8, line 162-167]
RBF interpolation for $f_j(\boldsymbol{\beta})$ can be expressed by the following equation:
$$f(\boldsymbol{\beta}) = \sum_{i=1}^{N} w_i \phi(\boldsymbol{\beta}, \boldsymbol{\beta}_i) = \exp(-\gamma||\boldsymbol{\beta} - \boldsymbol{\beta}_i||^2) \quad (17)$$
where $\boldsymbol{\beta}_i$ contains the set of parameters for case $i$, $w_i$ is the weight, and $\phi(\boldsymbol{\beta}, \boldsymbol{\beta}_i) \equiv \exp(-\gamma||\boldsymbol{\beta} - \boldsymbol{\beta}_i||^2)$ are RBF. Here, $\gamma$ is the parameter controlling the smoothness of the function. Also, by substituting the set of actual the input parameters in Eq. (17), we obtain the following simultaneous equations to determine the set of weights $w_i$:

$$\begin{pmatrix} f(\boldsymbol{\beta}_1) \\ \vdots \\ f(\boldsymbol{\beta}_N) \end{pmatrix} = \begin{pmatrix} \phi(\boldsymbol{\beta}_1, \boldsymbol{\beta}_1) & \cdots & \phi(\boldsymbol{\beta}_1, \boldsymbol{\beta}_N) \\ \vdots & & \vdots \\ \phi(\boldsymbol{\beta}_N, \boldsymbol{\beta}_1) & \cdots & \phi(\boldsymbol{\beta}_N, \boldsymbol{\beta}_N) \end{pmatrix} \begin{pmatrix} w_1 \\ \vdots \\ w_N \end{pmatrix} \quad (18)$$

[Revised manuscript, page 8, line 171-177]
RBF interpolation for $f_k(\boldsymbol{\beta})$ can be expressed by the following equation:
$$f_k(\boldsymbol{\beta}) = \sum_{i=1}^{N} w_i \phi(\boldsymbol{\beta}, \boldsymbol{\beta}_i) = \sum_{i=1}^{N} w_i \exp(-\gamma||\boldsymbol{\beta} - \boldsymbol{\beta}_i||^2) \quad (k = 1, ..., r) \quad (17)$$
where $\boldsymbol{\beta}_i$ contains the set of parameters for case $i$, $w_i$ is the weight, and $\phi(\boldsymbol{\beta}, \boldsymbol{\beta}_i) \equiv \exp(-\gamma||\boldsymbol{\beta} - \boldsymbol{\beta}_i||^2)$ are RBF. Here, $\gamma$ is the parameter controlling the smoothness of the function. Because $f_k(\boldsymbol{\beta})$ that is a coefficient of mode $k$ shown in Eq. (17) is individually interpolated, $w_i$ and $\gamma$ have different values for each mode. By substituting the set of actual the input parameters in Eq. (17), we obtain the following simultaneous equations to determine the set of weights $w_i$:

$$\begin{pmatrix} \alpha_{1k} \\ \vdots \\ \alpha_{Nk} \end{pmatrix} = \begin{pmatrix} \phi(\boldsymbol{\beta}_1, \boldsymbol{\beta}_1) & \cdots & \phi(\boldsymbol{\beta}_1, \boldsymbol{\beta}_N) \\ \vdots & & \vdots \\ \phi(\boldsymbol{\beta}_N, \boldsymbol{\beta}_1) & \cdots & \phi(\boldsymbol{\beta}_N, \boldsymbol{\beta}_N) \end{pmatrix} \begin{pmatrix} w_1 \\ \vdots \\ w_N \end{pmatrix} \quad (k = 1, ..., r) \quad (18)$$

CC 1-9

> Lines 188-190 weren't clear to me. Maybe say the slip was varied from 0.7 to 1.4 times the original slip in the model of Figure 2. When I first read this I also thought you were varying the rake between -20 and +25 degrees, which would be wrong for a subduction event, so maybe also make it even clearer that these are the range of perturbations in the rake angle from the ones given by Fujii-Satake?

We determined the ranges of the fault parameters by following the calculation condition reported by Kotani et al. (2020). According to the paper, JSCE (Japan Society of Civil Engineers (2011, in Japanese) conducted calculations with different rake values and reported that $\pm 10$ is a suitable rake range. But Kotani et al. (2020) changed the rake from -20 and +25 degrees to cover a more suitable range and check the effect of the variation of rake. Hence, the range of the rake used in our study may not be based on the real perturbations suggested in the Fujii-Satake model. Nevertheless, because the main objective of our study is to propose the instant prediction of tsunami forces, we think the wider range of the rake is

not necessarily an irrelevant condition. We ask for the reviewer's understanding.

CC 1-10

In Figure 5 the cyan lines for the Obs. are very hard to see, maybe make these lines black or red?

As the reviewer suggested, we have changed colors of these lines in red and black.

[Original manuscript, page 11, Figure 5]
Figure 5. Comparison between observational data and simulation results (adapted from Kotani et al. (2020)).
[Revised manuscript, page 12, Figure 5]
Figure 5. Comparison of tsunami heights between observational data and simulation results. (borrowed from Kotani et al. (2020))

CC 1-11

Line 209, by "concrete connection method" I think you mean the method for coupling the 2D and 3D methods together, but this was confusing to me at first. Maybe say something like "To couple the 2D and 3D models together, the method used in the study of Takase et al (2016) was employed." (By "concrete" I think you mean the specific method employed here, and it might be better to use "specific" here and some other places. Since in English concrete is also a building material, and you are talking about forces on buildings, there could be some confusion.)

Certainly, "concrete" also means a building material and so may be not appropriate here as an adjective. We have replaced "concrete" by "specific" as suggested by reviewer. We also applied same revisions in some other parts.

[Original manuscript, page 8, line 184-186]
In concrete terms, in our study we used a fault model (Fujii-Satake model Ver. 8.0) (Satake et al. (2013)) composed of the 55 small faults shown in Fig.2.
[Revised manuscript, page 8, line 194-196]
In specific terms, in our study we used a fault model (Fujii-Satake model Ver. 8.0) (Satake et al. (2013)) composed of the 55 small faults shown in Fig.2.

[Original manuscript, page 12, line 208-210]
Using the results of the 2D tsunami analysis over a wide area as the input conditions, a 3D tsunami analysis was performed to represent the tsunami runup in the target region. In terms of the concrete connection method, the method used in the study of Takase et al. (2016) was employed.
[Revised manuscript, page 11, line 218-220]
Using the results of the 2D tsunami analysis over a wide area as the input conditions, a 3D tsunami analysis was performed to represent the tsunami runup in the target region. As mentioned before, the method used in Takase et al. (2016) was employed to couple the 2D and 3D models together.

[Original manuscript, page 16, line 250-251]

Additionally, as a concrete parameter, $U_i$ expresses slip in relation to scenario (slip), and $\lambda_i$ expresses rake in relation to scenario $i$.

[Revised manuscript, page 16, line 267-268]

Additionally, as a specific parameter, $U_i$ expresses slip in relation to scenario (slip), and $\lambda_i$ expresses rake in relation to scenario $i$.

[Original manuscript, page 18, line 283]

In concrete terms, this coefficient can be expressed as $f_k(U, \lambda, t)$ as in the following equation.

[Revised manuscript, page 19, line 303]

In specific terms, this coefficient can be expressed as $f_k(U, \lambda, t)$ as in the following equation.

[Original manuscript, page 30, line 309]

In concrete terms, the error rate is calculated.

[Revised manuscript, page 25, line 329]

In specific terms, the error rate is calculated.
* * *
**CC 1-12**

> Line 231, discussing the 2D mesh used for evaluating the tsunami force: do you average (or sum?) the force over all vertical building faces that happen to lie in the 10m cell? It seems like this would vary a lot from cell to cell just based on the particular geometry of the buildings. In particular some cells might include no walls and hence have 0 force (?) while neighboring cells might have one wall or perhaps at the corner of a building a cell has two walls. So I'm surprised the plots of forces look as smooth as they do and perhaps you can say more about this?

The force is calculated by integrating the pressure acting on all the vertical faces of building within each evaluation cell. Therefore, as the reviewer pointed out, the tsunami force strongly depends on the surface area of buildings, and cells with no walls have zero force. Although we might have been able to employ other risk indicators, such as average pressure, the force was considered to be the simplest to check the result. Nevertheless, further insight into this aspect is left to future work.
* * *
**CC 1-13**

> It is great that you can get a surrogate model that reproduces the time evolution as well as it does, in addition to the spatial patterns. But I wonder if this is mainly because you are only considering perturbations to the Fujii-Satake model in which the basic spatial structure of the fault slip is always the same and so the time evolution shows similar sets of waves and arrival times, just somewhat varying magnitudes? The results are still impressive, but I wonder if you can comment on how this might be extended to developing a surrogate model that could be useful in real time for some earthquake that is not a small perturbation of 2011 (which the next big one almost certainly won't be).

As the reviewer pointed out, one of the reason why we could reproduce the time evolution using spatial modes would be that the basic spatial structure of the fault is always the same. Although we determined the ranges of the fault parameters by following Kotani et al. (2020), as the reviewer suggested, we might be able to check the potential of the surrogate models by considering wider range of the slip. Basically, the surrogate models can work even under large perturbation if enough calculation cases are available. But, even in that case, we have to consider the effective parameter sampling. This remains to be explored in future studies. While the point had been explained in conclusion, we didn't mentioned about parameter sampling. Therefore, we have added the view point in the conclusions.

[Original manuscript, page 31, line 347-349]

However, when increasing the types of input parameters, as the number of scenarios that need to be considered is immense, it is important to use an efficient method for setting the scenarios (McKay (1979)). Furthermore, it is necessary to evaluate in advance the extent to which these uncertainty parameters will fluctuate.

[Revised manuscript, page 27, line 368-370]

However, when increasing the types of input parameters, as the number of scenarios that need to be considered is immense, it is important to use an efficient method such as parameter sampling (McKay (1979)) for setting the scenarios. Furthermore, it is very important to evaluate in advance the extent to which these uncertainty parameters will fluctuate.
* * *
**CC 1-14**

> To develop a surrogate model that would handle a greater variety of quakes, perhaps it would be necessary to give up on trying to model the full time-dependent solution and instead build a surrogate model that only attempts to predict the maximum inundation depth and force, which might be much easier to do and still very useful.

As the reviewer suggested, it might be better to construct the surrogate models for only the maximum values in a greater variety of scenarios. But if we can consider both the space and time as in the manuscript, the resulting surrogate model can contain various information and of course accommodates the distribution of the maximum value. We believe there is still room for the time-space surrogate modeling even in a greater variety of scenarios. Please allow us to discuss the point in our future works.
* * *
**CC 1-15**

> I don't understand some of the discussion in the paragraph just below Table 2 (lines 312-320). You say "the ratio of mean values was 434%...". I think a ratio should just be a number, not a percent. Do you mean the ratio was 4.34? And what mean values is this the ratio of? Is it the ratio of the error to the true result, i.e. the relative error? This isn't clear.
>
> Maybe this would be clearer if you included also tables of the raw numbers you are comparing, it's not clear where these numbers come from or how they relate to Table 2.
>
> In line 316-317 you give ratios like 0.78%. Again I'm not sure what you mean by a ratio as a

> percent, do you mean the ratio of error to true value is 0.0078?

As the reviewer pointed out, the explanation of the error indicated in Subsection 3.2.4 was not appropriate. It might be difficult to understand. We have modified the expressions and Table 2.

[Original manuscript, page 30, line 309-310]
For each of the physical quantities, the mean squared error for the time-series data of all points are calculated. The results are shown as follows in Table 2.
[Revised manuscript, page 25, line 329-330]
For each of the physical quantities, the root mean squared error (RMSE) for the time-series data of all points are calculated by using Eq. (21). The results are shown as follows in Table 2.

[Original manuscript, page 30, line 311-312]
With regard to Table 2, the ratio of the mean values was 434% for the impact force data and 414% for the water depth data in S2R3 and 318% for the impact force data and 252% for the water depth data in S5R7.
[Revised manuscript, page 25, line 331-333]
With regard to Table 2, the relative error to the mean value of time series data for a specific scenario was 434% for the impact force data and 414% for the water depth data in S2R3 and 318% for the impact force data and 252% for the water depth data in S5R7.

[Original manuscript, page 30, line 316-318]
Note that the ratio of the maximum values is 0.78% for the impact force data and 2.15% for the water depth data for S2R3 and 0.78% for the impact force data and 2.04% for the water depth data in the case of S5R7.
[Revised manuscript, page 25-26, line 337-339]
Note that the relative error to the maximum value of time series data for a specific scenario is 0.78% for the impact force data and 2.15% for the water depth data for S2R3 and 0.78% for the impact force data and 2.04% for the water depth data in the case of S5R7.

[Original manuscript, page 30, Table 2]

| case name | error [N] (impact force) | error [m] (inundation) |
| --- | --- | --- |
| S2R3 | 66930 | 0.326 |
| S5R7 | 84017 | 0.404 |

[Revised manuscript, page 26, Table 2]

| Case name | S2R3 | | S5R7 | |
|---|---|---|---|---|
| | Impact force | Inundation | Impact force | Inundation |
| Error (RMSE) | $6.69 \times 10^4$ [N] | 0.326 [m] | $8.40 \times 10^4$ [N] | 0.404 [m] |
| Mean value | $1.54 \times 10^4$ [N] | 0.0787 [m] | $2.64 \times 10^4$ [N] | 0.160 [m] |
| (RMSE)/(Mean value) | 434 [%] | 414 [%] | 318 [%] | 252 [%] |
| Maximum value | $8.59 \times 10^6$ [N] | 15.1 [m] | $1.08 \times 10^7$ [N] | 19.8 [m] |
| (RMSE)/(Mean value) | 0.78 [%] | 2.15 [%] | 0.78 [%] | 2.04 [%] |

**CC 1-16**

> In spite of my questions and possible confusion, in general I like the paper and believe it should be published after some clarifications.

We thank the reviewer for taking time to review our manuscript with a lot of valuable suggestions and appropriate advice. Thanks to the reviewer's comments and suggestions, the quality of our paper have improved significantly.

**#RC3 2021/9/22**

RC 3-1

> The authors tried to estimate the hydrodynamic force of tsunami acting on building through 2D and 3D FEM. The paper is interesting, and from my personal point of view, the topic of this manuscript fits well with the scope of the journal of NHESS. There are too many figures (snapshots), however few explanations and not ready. Animations or supplementary files may be attached to improve readability. Some other major issues are listed below:

We appreciate your valuable suggestion. We have provided animations that visualize the results obtained in a calculation case (S3R5).

RC 3-2

> 1. One faulting model of Tohoku Earthquake 2011 is used in this study, however but it may not (and should not) be reactivate at the same location and did not have the same magnitude. So it cannot be applied to other events, nor can it "predict" damages. Therefore, "real-time" tsunami force predictions may not be useful because the events of 2011 have already occurred. The term "real time" may change with the reassessment of the power of the tsunami...
>
> 2. The method proposed by the authors is a time-consuming task and not taking into account the faulting parameter of individual event. Likewise, the term "real-time" may not be a suitable term.

As the reviewer pointed out, it takes much time to perform numerical simulations with a lot of scenarios. However, all the simulations are conducted in advance to construct a surrogate model. Once the model is completed, the real-time prediction is ready. That is, when an actual tsunami occurs and its fault parameters are determined in some way, the present surrogate model can predict not only inundation heights and areas, but also hydrodynamic forces of tsunami acting on buildings within a very short period of time. Although we applied the proposed method to the Tohoku Eearthquake 2011 in this study, the framework has broad utility and therefore is capable of estimating risks of other events.

Nevertheless, we have recognized that the term "real-time" may not be suitable. Indeed, similar comments have been left by other reviewers. In response, we have replaced "real-time" by "Rapid" in the manuscript including the title. The revised parts are shown in the response to RC 2-1.

RC 3-3

> 3. There are too many figures, but similar···, need to be simplified. The wording of the manuscript should be further elaborated so that the reader can understand what the author wants to express, regarding to the figures.
>
> 4. Many figures is not ready, and some parts can be unified. E.g. Fig 6, 7, 8, 11, 17 etc.

As you pointed out, there may be unnecessarily many pictures and some of them can be simplified. Also, there was some insufficient captions. Similar comments were also provided by the other reviewer (RC 2-11). The merged and revised figures are shown in the response to RC 2-11.

---

## Author Response (AR2)

**Responses to review comments**

We deeply appreciate reviewers' valuable comments and kind suggestions. We have revised the manuscript accordingly and present responses to them in this sheet. The revised parts are colored in red in the following responses and the revised manuscript.

**The main points:**

1. The literature review is inadequate.

There are too few references on the relevant past work on surrogate modelling (also known as emulation or meta-modelling or response surface). Please cite several papers (say 5-6?) with various settings (e.g. parameters ranges of earthquakes/landslides sources or other uncertain parameters such as roughness, applications to faster than real time early warning or hazard assessments) for tsunami emulation. There are papers on this topic in NHESS and other journals in the last five years or so, I let the authors decide which ones to cite and comment on to contrast with their approach. Please discuss them a bit to help the reader situate your work. In particular, the mainstream method of Gaussian Process (GP) emulation to create surrogates is powerful as it provides uncertainties and allows for high nonlinearities, so must be mentioned and discussed, as these generalise in some way equation (17) in this paper.

Remove the sentence and references "LeVeque et al. (2016) and Melgar et al. (2016) utilize the Karhunen–Loève expansion to consider the distribution of slips on a fault under various scenarios" as these are irrelevant here and the sentence is out of place.

The other references are relevant and fine to keep.

As the reviewer pointed out, the explanations of past studies on surrogate modeling may have not been enough. Thus, we have added the explanations in the introduction to clarify the difference from them. Since the RBF interpolation is regarded as a Gaussian Process emulation, we have added this point in the manuscript. In addition, we have removed the sentence "LeVeque et al. (2016) and Melgar et al. (2016) utilize the Karhunen–Loève expansion to consider the distribution of slips on a fault under various scenarios", because it was irrelevant to this part, as you pointed out.

**[Original manuscript, Page 2, Line 40-43]**

Therefore, a surrogate modeling-based prediction method is employed in this study. However, since this study considers multiple evaluation points to determine tsunami force, the methods employed in Fukutani et al. (2019) and Kotani et al. (2020), which require surrogate models to be defined at each evaluation point, are inefficient. To overcome this problem, a mode decomposition technique is used to construct surrogate models.

**[Revised manuscript, Page 2, Line 41-48]**

Therefore, a surrogate modeling-based prediction method is employed in this study. Surrogate modeling has been widely accepted for uncertainty quantification and probabilistic risk assessment, such as studies using Response Surface (e.g., Fukutani et al., 2019; Kotani et al., 2020), Gaussian Process (e.g., Sarri et al., 2012; Salmanidou et al., 2017, 2021), Polynomial Chaos Expansion (e.g., Denamiel et al., 2019; Giraldi et al., 2017; Sraj et al., 2017), and Multifidelity Sparse Grids (e.g., de Baar and Roberts, 2017). These works demonstrate the potential of the Surrogate modeling-based approach, while the surrogate model considering spatio-temporal variation has not been well studied. Since this study considers time variation of spatial distribution of tsunami risk, a mode decomposition technique is efficiently used to construct surrogate models.

**[Original manuscript, Page 7, Line 169-170]**

In this study,  $f_k(\beta)$  is determined by the interpolation with radial basis functions (RBF) (Buhmann, 1990) because it is applicable even if the parameters are not distributed in equidistant intervals and even if some defects are present in the data.

**[Revised manuscript, Page 7, Line 172-174]**

In this study,  $f_k(\beta)$  is determined by the interpolation with radial basis functions (RBF) interpolation (Buhmann, 1990) which is regarded as a Gaussian Process (GP) regression. The GP regression is applicable even if the parameters are not distributed in equidistant intervals and even if some defects and high non-linearity are present in the data.

**[Original manuscript, Page 2, Line 47-48]**

LeVeque et al. (2016) and Melgar et al. (2016) utilize the Karhunen–Loève expansion to consider the distribution of slips on a fault under various scenarios.

【Revised manuscript】 (Removed)

2. Section 2.3 needs a bit of clarification. line 167 should be "input parameters beta:". The error in equation (18) is not mentioned, but it is present as it is a regression equation and should be in the

equation. Line 1801-182 more details are needed on what cross-validation is used, why a Bayesian optimisation if used and how.

As the reviewer pointed out,  $f_k(\beta)$  in line 167 was a mistake, we have fixed it in the manuscript.

[Original manuscript, Page 7, Line 166-167]

Then, the surrogate model can be expressed in the following equation whose independent variables are input parameters  $f_k(\boldsymbol{\beta})$ :

[Revised manuscript, Page 7, Line 169-170]

Then, the surrogate model can be expressed in the following equation whose independent variables are input parameters  $\beta$ :

Because the explanation of regression was insufficient, we have added an explanation including the regression equation. In addition, since there were no detailed explanations of cross-validation and Bayesian optimization in the manuscript, we have added their explanations.

[Original manuscript, Page 8, Line 178-182]

In this study, ridge regression (Hoerl and Kennard, 1970) is used to compute weights for Eq. (18). By introducing the ridge regression, it is possible to prevent the overfitting problem. Since the accuracy of the interpolation depends on the regularization parameter used in the ridge regression and the smoothness parameter of the RBF interpolation, these values are determined by the application of a certain cross-validation technique (Stone, 1974) combined with the Bayesian optimization (Mockus, 1975) in this study.

[Revised manuscript, Page 8, Line 182-190]

In this study, ridge regression (Hoerl and Kennard, 1970) is used to compute weights for Eq. (18). The weights are obtained by solving the following optimization problem:

$$\arg\min_{\mathbf{w}_{k}}\left(\left|\left|\boldsymbol{\alpha}_{k}-\boldsymbol{\Phi}\mathbf{w}_{k}\right|\right|_{2}^{2}+\lambda\left|\left|\mathbf{w}_{k}\right|\right|_{2}^{2}\right) \quad (k=1,\ldots,r)$$
(19)

where  $\lambda$  is the regularization parameter,  $\alpha_k$  is the vector of coefficients for the *k*-th mode,  $w_k$  is the vector of weights for the *k*-th mode, and  $\Phi$  is the coefficient matrix in equation (18). By introducing the ridge regression, it is possible to prevent the overfitting problem. Since the accuracy of the interpolation depends on the regularization parameter  $\lambda$  and the smoothness parameter  $\gamma$  of the RBF interpolation, these values are determined by the application of a certain cross-validation technique (Stone, 1974). In this study, the 4-folded cross-validation is applied to determine the parameters  $\gamma$  and  $\lambda$ , and the Bayesian optimization (Mockus, 1975) is used to efficiently search for the optimal values of  $\gamma$  and  $\lambda$  in the parameter space.

3. Table 1 and 4. Conclusion: The design used is a grid, which is much less efficient than Latin Hybercubes or sequential designs for GP emulation, or sparse designs for Polynomial Chaos surrogates: nobody uses a grid anymore to construct emulators. A reference to more efficient methods, ideally in tsunami emulation/surrogate context is needed. The study would have greatly benefited from a better design using the same computational budget.

This study mainly presents the framework of the surrogate model using the mode decomposition technique and does not focus on the sampling of simulation scenarios. However, as the reviewer suggested, Gaussian Process, Polynomial Chaos, and other sampling methods can also be applied in the framework, and combining with such methods can probably improve accuracy of the surrogate model. In order to mention this point, we have added an explanation in the conclusion.

[Original manuscript, Page 27, Line 368-369]

However, when increasing the types of input parameters, as the number of scenarios that need to be considered is immense, it is important to use an efficient method such as parameter sampling (McKay (1979)) for setting the scenarios.

**[Revised manuscript, Page 28, Line 381-383]**

However, when increasing the types of input parameters, as the number of scenarios that need to be considered becomes immense, it is pertinent to use efficient methods for parameter sampling such as Latin Hypercubes (McKay (1979)), etc.

4. Performance is explored on the 10 cases of  $+_10$  deg rake. This are quite central so may not represent the whole range of inadequacies of the surrogate. Usually a leave-one-out diagnostic is performed and would not cost much here as there is no need for any more simulation. It would show a better picture than this narrow set of scenarios that might be rather easy to predict.

As the reviewer pointed out, there is room for more detailed validation. However, the main objective of this study is to propose a framework for rapid tsunami force prediction. We believe that the present

surrogate models have enough performance for the purpose of demonstrating the effectiveness of the framework. Because the accuracy of the surrogate models should be discussed in conjunction with the sophistication of parametric sampling, it would be helpful if we could make this as one of the future works. We ask for the reviewer's understanding.

5. Please discuss more the sometimes 50% differences between model and surrogate outputs in fig 18-19: these could be very local aspects of topography that cannot be captured by a global POD approach that assesses the overall variability. This is one major deficiency in this application, so needs to be acknowledged. Please check the relative errors of 400% as they do not seem to match the figures: these should be around 50% max.

As the reviewer pointed out, the large error reflects the local aspects of topography and is an important point we should notice. Therefore, we have added an explanation to clarify the point. Also, in response to this point reviewer raised, we have checked again the large relative error using the mean value and then found that the calculation was incorrect. Accordingly, we have revised the relative errors in Table 2 and in the manuscript.

**[Original manuscript, Page 25, Line 328-333]**

Here, it is confirmed that it is possible to represent the simulation results using arbitrary parameters with the surrogate model. In specific terms, the error rate is calculated. For each of the physical quantities, the root mean squared error (RMSE) for the time-series data of all points are calculated by using Eq. (21). The results are shown as follows in Table 2.

With regard to Table 2, the relative error to the mean value of time series data for a specific scenario was 434% for the impact force data and 414% for the water depth data in S2R3 and 318% for the impact force data and 252% for the water depth data in S5R7.

**[Revised manuscript, Page 27, Line 337-346]**

Here, it is confirmed that it is possible to represent the simulation results using arbitrary parameters with the surrogate model. However, as shown in Figs. 18 and 19, it should be noted that the surrogate model does not fully represent the local spatial distribution. More advanced mode decomposition techniques, such as the sparse modeling, are needed to improve the accuracy.

Next, the root-mean-squared error (RMSE) for the time-series data at all points are calculated for each of the physical quantities by using Eq. (22). The relative errors to the mean value and the maximum value are presented in Table 2. Here, "Mean value" is a root-mean-square of the

**corresponding quantity.**

With regard to Table 2, the relative error to the mean value of time series data for a specific scenario was 37.2% for the impact force data and 45.1% for the water depth data in S2R3 and 32.4% for the impact force data and 36.7% for the water depth data in S5R7.

| Case name           | S2R3                      |                | S5R7                      |            |
|---------------------|---------------------------|----------------|---------------------------|------------|
|                     | Impact force              | Inundation     | Impact force              | Inundation |
| Error (RMSE)        | $6.69\times10^4~[\rm N]$  | $0.326 \ [m]$  | $8.40 \times 10^4  [N]$   | 0.404 [m]  |
| Mean value          | $1.54 \times 10^4  [N]$   | $0.0787 \ [m]$ | $2.64 \times 10^4 \; [N]$ | 0.160 [m]  |
| (RMSE)/(Mean value) | 434 [%]                   | 414 [%]        | 318 [%]                   | 252~[%]    |
| Maximum value       | $8.59\times 10^6~[\rm N]$ | 15.1 [m]       | $1.08\times 10^7~[\rm N]$ | 19.8 [m]   |
| (RMSE)/(Mean value) | 0.78~[%]                  | 2.15~[%]       | 0.78~[%]                  | 2.04 [%]   |

**[Original manuscript, Page 26, Table 2]**

**[Revised manuscript, Page 27, Table 2]**

| Case name                   | S2R3                      |               | S5R7                      |               |
|-----------------------------|---------------------------|---------------|---------------------------|---------------|
|                             | Impact force              | Inundation    | Impact force              | Inundation    |
| Error (RMSE, Equation (22)) | $6.69 \times 10^4 \; [N]$ | $0.326 \ [m]$ | $8.40 \times 10^4 \; [N]$ | $0.404 \ [m]$ |
| Mean value                  | $1.80 \times 10^5$ [N]    | $0.722 \ [m]$ | $2.59\times10^5~[\rm N]$  | 1.10 [m]      |
| (RMSE)/(Mean value)         | 37.2 [%]                  | 45.1 [%]      | 32.4~[%]                  | 36.7~[%]      |
| Maximum value               | $8.59 \times 10^6  [N]$   | $15.1 \; [m]$ | $1.08 \times 10^7  [N]$   | 19.8 [m]      |
| (RMSE)/(Maximum value)      | 0.78 [%]                  | 2.15~[%]      | 0.78 [%]                  | 2.04 [%]      |

**Minor points:**

1. how do you extract spatial modes in Fig 11 across times and temporal modes in line 274? A bit more clarity about the role of time is needed in the text.

As shown in Equation 20, all spatial distributions obtained in the simulation are included in the data matrix. This means that the spatial modes are common to all time and all scenarios. Temporal effects are expressed only by the coefficients. To make the explanation clearer, we have added an explanation.

**[Original manuscript, Page 16, Line 274-275]**

By applying proper orthogonal decomposition to this matrix, it is possible to extract a common

temporal mode for data including time, and it is possible to construct a surrogate model including the time direction.

**[Revised manuscript, Page 17, Line 282-284]**

By applying proper orthogonal decomposition to this matrix, it is possible to extract a common temporal mode for data including time. Temporal effects are expressed by the coefficients of the modes, and it is possible to construct a surrogate model including the time direction.

**Other revision**

Because an article that should be cited in this manuscript was published during the peer review period, we have newly cited it in Introduction.

**[Original manuscript, Page 2, Line 34-37]**

Among these is a study on making real-time predictions using source information and precomputed tsunami waveform and inundation databases (Gusman et al., 2014), another uses a combination of precomputed tsunami databases and the neural networks (Fauzi and Mizutani, 2019), and another features the surrogate modeling of numerical simulation results (e.g., Fukutani et al., 2019; Kotani et al., 2020).

**[Revised manuscript, Page 2, Line 34-38]**

Among these is a study on making real-time predictions using source information and precomputed tsunami waveform and inundation databases (Gusman et al., 2014), another uses a combination of precomputed tsunami databases and the neural networks (Fauzi and Mizutani, 2019; Liu et al., 2021), and another features the surrogate modeling of numerical simulation results (e.g., Fukutani et al., 2019; Kotani et al., 2020).